



# Gas-Phase Pyrolysis Products Emitted by Prescribed Fires in Pine Forests with a Shrub Understory in the Southeastern United States[**]

Nicole K. Scharko[1], Ashley M. Oeck[1], Tanya L. Myers[1], Russell G. Tonkyn[1],
Catherine A. Banach[1], Stephen P. Baker[2], Emily N. Lincoln[2], Joey Chong[3],
Bonni M. Corcoran[3], Gloria M. Burke[3], Roger D. Ottmar[4], Joseph C. Restaino[5],
David R. Weise[3], and Timothy J. Johnson[1*]

[1]Pacific Northwest National Laboratories, Richland, WA, USA
[2]USDA Forest Service, Rocky Mountain Research Station, Missoula, MT, USA
[3]USDA Forest Service, Pacific Southwest Research Station, Riverside, CA, USA
[4]USDA Forest Service, Pacific Northwest Research Station, Seattle WA, USA
[5]School of Environmental and Forest Sciences, University of Washington, Seattle WA, USA

*To whom correspondence should be addressed: Timothy.Johnson@pnnl.gov

## ABSTRACT

In this study we capture and identify pyrolysis gases from prescribed burns conducted in pine forests with a shrub understory using a manual extraction device. The device selectively sampled emissions ahead of the flame front, minimizing collection of oxidized gases, with the captured gases analyzed in the laboratory using infrared absorption spectroscopy. Results show that emission ratios (ER) relative to CO for ethene, and acetylene were significantly greater than previous fire studies, suggesting that the sample device was able to collect gases prior to ignition. Further evidence that ignition had not begun was corroborated by novel infrared detections of several species, in particular naphthalene. With regards to oxygenated species, several aldehydes (acrolein, furaldehyde, acetaldehyde, formaldehyde) and the carboxylic acids (formic, acetic) were all observed; results show that ERs for acetaldehyde were noticeably greater while ERs for formaldehyde and acetic acid were lower compared to other studies. The acetylene-to-furan ratio also suggests that high temperature pyrolysis was the dominant process generating the collected gases. This hypothesis is further supported by the presence of HCN and the absence of $NH_3$.



[**]This manuscript was prepared, in part, by a U.S. Government employee on official time, is not subject to copyright and is in the public domain. The use of trade or firm names in this publication is for reader information and does not imply endorsement by the U.S. Department of Agriculture of any product or service.

## 1. INTRODUCTION

Biomass burning contributes large quantities of trace gases into the earth's atmosphere (Crutzen and Andreae, 1990; Akagi et al., 2011; Andreae and Merlet, 2001; Crutzen et al., 1979; Yokelson et al., 2013; Andreae, 1991). The primary carbon-containing gases emitted during such burns are $CO_2$, CO and $CH_4$, in order of decreasing concentration (Ward and Hardy, 1991). Hundreds of other trace gases have also been identified in the emissions, including many non-methane volatile organic compounds (NMVOCs), oxygenated volatile organic compounds (OVOCs), nitrogen-containing species and sulfur compounds (Yokelson et al., 1996; Lobert et al., 1991; Talbot et al., 1988). The major sources of biomass burn emissions are wildland fire and, to a lesser extent, prescribed fire. Prescribed fires are used by foresters, other officials and private land owners to reduce dangerous fuel buildups and manage habitats (Fernandes and Botelho, 2003). The use of prescribed fire as a preventative tool is of particular importance in the western United States (U.S.) where wildland fires are increasing in severity (Turetsky et al., 2011; Miller et al., 2009). In the southeastern U.S., prescribed fire is also used on a routine basis for purposes such as ecosystem management (Waldrop and Goodrick, 2012). For these and other beneficial reasons, an estimated 3.6 million hectares of forestry land have been burned in the U.S. by prescribed fire each year (Melvin, 2012). Agencies that conduct such burns often rely on fire-related models (Reinhardt et al., 1997; Prichard et al., 2006) to predict the impacts of the prescribed burn. Having detailed knowledge of the emission products can thus improve the predictive output of such regional models (Reid et al., 2009).

Due to the influential role of wildland fire on atmospheric chemistry and climate, there has been considerable interest in identifying and quantifying gas emissions from fire as studied both in the





laboratory and in field burns (Crutzen et al., 1979; Andreae et al., 1988; Lobert et al., 1991;
Andreae et al., 1994; Lindesay et al., 1996; Goode et al., 1999; Yokelson et al., 1999; Yokelson et
al., 1996; Chi et al., 1979). The type of gases emitted and their relative abundances depend on
many factors such as fuel type, fuel arrangement, land management activities, burning techniques
and environmental conditions (Ward et al., 1996; Ward et al., 1992). In the 1990s, Griffith,
Yokelson and co-workers conducted a series of laboratory studies using an open-path Fourier
transform infrared (FTIR) spectrometer to investigate how some of these factors influence the
emitted gases (Goode et al., 1999; Yokelson et al., 1996; Yokelson et al., 1997). There have been
several follow-on laboratory studies using IR spectroscopy along with other analytical techniques
to identify previously unknown fire emission products and to derive emission factors from various
fuel types (Burling et al., 2010; Hatch et al., 2017; Selimovic et al., 2018; Stockwell et al., 2014;
Yokelson et al., 2013; Gilman et al., 2015).

In addition to the laboratory studies, a number of field campaigns have also used FTIR
spectroscopy to identify trace gases from prescribed fires (Akagi et al., 2013; Burling et al., 2011;
Akagi et al., 2014; Goode et al., 2000; Yokelson et al., 1999; Wooster et al., 2011; Alves et al.,
2010; Hurst et al., 1994a; Hurst et al., 1994b; Paton-Walsh et al., 2010; Paton-Walsh et al., 2008;
Guérette et al., 2018). Studies that have the ability to measure emissions both near the fire and
aloft are especially useful in understanding the complex chemistries that occur during and after
prescribed fires, including the (oxidative) chemistry of the downwind plume. For example, (Akagi
et al., 2013) detected limonene from a prescribed burn with a land-based FTIR and linked it to the
production of ozone, formaldehyde and methanol, all of which were measured downwind with an
airborne-based FTIR.  In an earlier prescribed burn study, Burling et al. (2011) detected enhanced

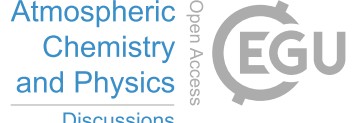



levels of isoprene and 1,3-butadiene in the smoke from a living tree when compared to dead stumps
under the same conditions. Emission characteristics obtained during such campaigns can be
especially useful for the implementation or verification of effective burning techniques to
minimize the gases released during prescribed burns.

However, few investigations have studied the pre-ignition or pyrolysis gases emitted prior to the
flaming combustion stage. Most prescribed burn studies have focused only on the flaming and
smoldering stages. The hotter flaming stage is characterized by more oxidized products and a
higher modified combustion efficiency (MCE) (Ward and Hao, 1991),  which is defined as:

$$\mathrm{MCE} = \left( \frac{\Delta CO_2}{\Delta CO_2 + \Delta CO} \right).$$
(1)

The cooler smoldering phase with lower MCE values (typically ranging from 0.65–0.85)
(Urbanski, 2013) displays more non-oxidized or less-oxidized species. The present study differs
from these earlier works in that we exclusively attempt to investigate pyrolysis, which is the first
step in the burning process (Collard et al., 2014), in particular we investigate the gas phase
pyrolysis species generated during prescribed burns.  Primary mechanisms associated with
pyrolysis of biomass are char formation, depolymerization and species fragmentation. Volatile
products are generated and, if unstable, can continue to undergo secondary (non-combustion)
reactions such as cracking or recombination (Collard and Blin, 2014).  Pyrolytic reactions produce
fuel gases that, if sufficient in quantity and in the presence of  oxygen, will maintain the flame via
combustion pathways (Ward and Hardy, 1991; Di Blasi, 1993). Thus, the primary objectives of
the present study are a) to detect pyrolysis gases in prescribed burns (i.e. gases that are emitted
prior to the flame front and prior to the onset of combustion) and b) to determine if they are
different from pyrolysis gases measured under more tightly controlled laboratory conditions.




There have been several pyrolysis laboratory studies carried out in controlled environments: In one of the earliest investigations, DeGroot et al. (1988) detected $H_2O$, $CO_2$, $CH_3OH$, HCOOH and $CH_3COOH$ from the pyrolysis of wood. More recent studies have observed several other compounds, such as CO, $CH_4$, light weight hydrocarbons ($C_2$–$C_5$) and light tar compounds (e.g. benzene and its derivatives and polycyclic aromatic hydrocarbons) from the slow pyrolysis of Birch wood (Fagernäs et al., 2012). Oxygenated compounds (e.g. furan-related compounds) have been observed from the fast pyrolysis of levoglucosan, a known pyrolyzate of cellulose (Bai et al., 2013). Laboratory experiments that have investigated condensed and/or gas phase compounds generated by pyrolysis under controlled conditions have revealed that the speciation and distribution of the products are dependent on a number of factors such as heating rate, temperature, fuel composition, live vs. dead fuels and amount of available oxygen (Azeez et al., 2011; Lu et al., 2011; Shen et al., 2010; Safdari et al., 2018; Ren and Zhao, 2012, 2013a, b). For instance, Ren and coworkers (2013a) found that the amount and speciation of nitrogen containing pyrolyzates is complicated and influenced by the content of mineral matter, the presence of oxygen (Ren and Zhao, 2012), as well as the structure (e.g. aliphatic vs. heterocyclic) of the amino acids and the amount of cellulose, hemicellulose and lignin in the sample. Similarly, the release of oxygenated compounds (e.g. phenolic compounds) from the pyrolysis of lignin is sensitive to the presence of oxygen (Kibet et al., 2012). The pyrolysis studies mentioned above were conducted in controlled settings or on smaller scales. There remains a paucity of data that identify and quantify gas-phase pyrolysis species emitted from actual prescribed burns at the field scale.




To our knowledge, this is one of the first field studies that discriminatively measures pyrolysis
gases for southeastern U.S. fuels. Isolating such species is indeed challenging as they often blend
with the background atmosphere and are rapidly mixed with other gases at the onset of combustion.
One must thus isolate the "pyrolysis molecules" either optically, mechanically or temporally. In
this study, we selectively probe the pyrolysis gases by using a simple manually-operated spatial
collection device that attempts to ensure that only gases in front of the flame are collected. While
not a perfect solution, the information gathered in this study adds important insights into the
primary products generated during the pyrolysis process.

**2. EXPERIMENTAL**
**2.1 Site description**
During the week of 29 April 2018, a total of seven small plots (160 m$^2$) were burned using
prescribed fire at Ft. Jackson, South Carolina (SC), latitude: 34.05 and longitude: −80.83,
approximately 10 km east of Columbia, SC. The fort lies entirely within the Sandhills ecosystem
in the South Carolina coastal plain, which runs approximately parallel to the Atlantic Ocean coast,
175 km inland. The Sandhills region thus forms a belt that tracks southwest − northeast across
sands of varying depth with a high content of pure silica (Porcher and Rayner, 2001). The deep
sands support an overstory vegetation that has significant amounts of turkey oak (*Quercus laevis*
Walter) and two native pine species relatively unique to the southeastern U.S.: longleaf pine (*Pinus*
*palustris* Mill.) and slash pine (*Pinus elliottii* Engelm.). The understory has substantial quantities
of immature turkey oak, longleaf and slash pine, along with sparkleberry (*Vaccinium arboreum*
Marshall) and a heterogeneous organic layer of woody material, litter, duff and cones atop the
mineral soil. The longleaf ecosystem depends on fire for maintenance (Cary, 1932). Site details



for the seven burn plots, all with a 2 year rough (i.e. burned 2 years prior), are summarized in Table
1. Eight pre- and post-fire 1.0 meter square biomass clipped plots were established at each 160 m$^2$
research block where organic vegetative material in each site was collected before and after each
fire.  Shrub, grasses/forbs, down woody material (0-0.6, 0.6-2.5, 2.5-7.6 7.6-22.9 cm in diameter),
litter and duff, are the major fuel bed components that were targeted.  Fuel moisture samples for
each major component were collected before ignition to determine fuel moisture content for each
fuel bed component. Figure 1 shows photographs of site 16, plot 1 before, during and after the
burn as well as a thermal image of the flame interacting with the fuel.





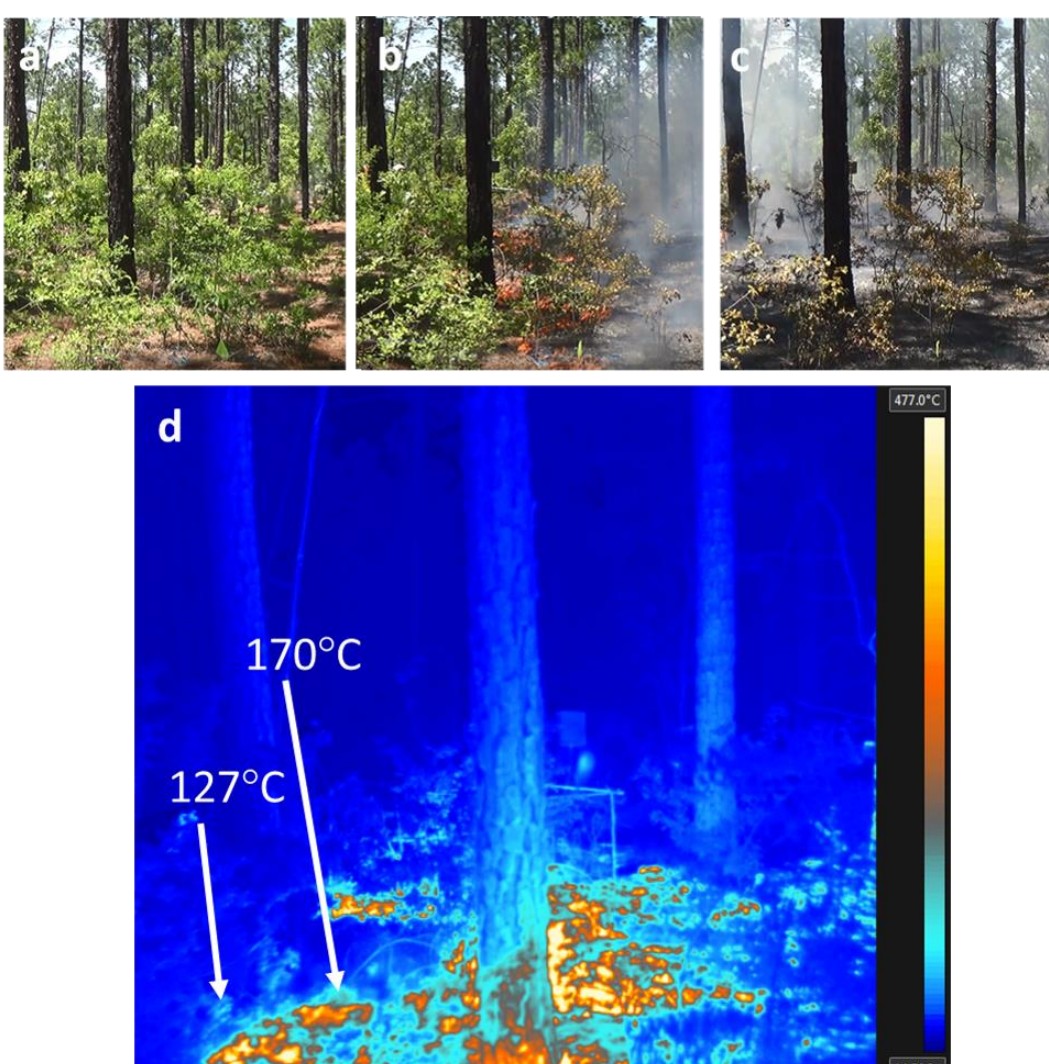

**Figure 1.** Photographs of site 16, plot 1 on 3 May 2018 between 14:00 and 14:40 local time. The plot (a) before the
flame, (b) while the flame interacted with the fuel at 14:33 and (c) smoldering combustion of the fuel. The primary
species seen in the understory are sparkleberry and a litter layer of pine needles. (d) Thermal image of the flame
interacting with the fuel at time of 14:33.



**Table 1.** Plot name, date/time, fuel description, atmospheric conditions (all clear sky days) and plot mixing height and area for the prescribed burns.

| Burn plot | Date (2018) | Local start time (EDT) | Local finish time (EDT) | Dominant overstory | Understory fuels | Ambient temperature (°C) | Relative humidity (%) | Surface winds ($m\,s^{-1}$) and wind direction | Mixing height (m) | Area burned ($m^2$) |
|---|---|---|---|---|---|---|---|---|---|---|
| 24B-triangle | 1-May | 12:11 | 12:37 | slash pine | sparkleberry/ logs | 24 | 26 | 2.7 SW | 975 | 450 |
| 24B-north diamond | 1-May | 13:53 | 14:43 | slash pine | sparkleberry/ logs | 28 | 18 | 2.7 SW | 1310 | 900 |
| 24A-square | 2-May | 9:37 | 10:22 | longleaf pine | sparkleberry/ duff | 21 | 53 | 2.7 SW | 792 | 900 |
| 24A-triangle | 2-May | 12:08 | 12:43 | longleaf pine | sparkleberry/ duff | 27 | 34 | 2.7 SW | 1189 | 450 |
| 16 plot 5 | 3-May | 9:39 | 10:21 | longleaf pine | sparkleberry/ bracken fern | 22 | 59 | 2.7 SW | 579 | 900 |
| 16 plot 6 | 3-May | 11:44 | 12:13 | longleaf pine | sparkleberry/ turkey oak | 26 | 43 | 3.1 SW | 1067 | 900 |
| 16 plot 1 | 3-May | 13:56 | 14:41 | longleaf pine | sparkleberry/ turkey oak | 29 | 30 | 3.1 SW | 1494 | 900 |

## 2.2 Collection device

Our approach to sampling used an extractive collection device whose tube inlet sampled air and emissions directly ahead of the flame. This simple solution is similar to other canister methods often used with gas chromatographic analysis (Young et al., 1997) and also conceptually similar to the land-based FTIR used to sample emissions as described by Akagi et al. (2013, 2014) and Burling et al. (2011). The canister sampling package, mounted on a metal frame, contained a set of evacuated canisters which were carried to the individual burn plots. The sampling package consisted of a 12-Volt Swing Piston KNF Neuberger Pump (NPK09DC) plumbed with stainless steel tubing to a pressure relief valve and gauge. The pressure relief valve was adjustable to regulate the pressure of the system and ultimately the fill pressure of the canisters. The flow rate to fill the canisters was 15 liters $min^{-1}$. A sampling probe (2.5 m of 6 mm stainless steel tubing plus 2 m of flexible stainless-steel line) was attached to the inlet of the package to collect pyrolysis gases from point sources of vegetation within the burning plots (Figure S1 displays a photo of the device). The device had an in-line two-way valve to control the sampling interval. To capture a



pyrolysis sample, the probe was placed near the base of the flame, immediately above the fuel
where the pyrolysis gases should be emitted at maximal levels. Seven to ten aliquots of gas sample
were added to a single canister as the device was moved in front of the flame to capture pyrolysis
gases. Each 3-liter Summa canister was filled to approximately 138 kPa (20 psia) for the FTIR
analysis.

**187 2.3 FTIR Spectrometer and Spectral Analysis**

The experimental details regarding FTIR measurement procedures have been previously reported
(Scharko et al., 2018). The FTIR spectrometer parameters and measurement details are briefly
summarized as is the spectral analysis: Gases in the canisters were returned from the field to the
laboratory and analyzed the same day or the following day using an 8 meter White cell (Bruker
Optics, A136/2-L) and FTIR; canisters were connected to the gas cell via 3/8" stainless steel tubing
with both the tubing and gas cell heated to 70 °C to prevent analyte adhesion.  The cell was coupled
to a purged FTIR (Bruker Optics, Tensor 37) spectrometer equipped with a glow bar source, KBr
beamsplitter and liquid nitrogen cooled mercury cadmium telluride (MCT) detector. Spectra were
collected from 4000 to 500 cm$^{-1}$ at 0.6 cm$^{-1}$ resolution.

Spectral analysis was carried out using the MALT5 program (Griffith, 2016)  and 50 °C reference
spectra from the PNNL database (Sharpe et al., 2004; Johnson et al., 2010) as well as absorption
lines from HITRAN (Gordon et al., 2017). MALT5 fits the assigned reference spectral lines to the
measured spectrum by optimizing the fit of all gases ascribed to the spectral window and
minimizing the residual. The calculation involves input parameters such as path length, resolution
and apodization accompanied by reference absorption cross-sections and the measured spectrum



with its associated temperature and pressure values. Both $H_2O$ and $CO_2$ had peaks that were
optically saturated; these regions were eliminated from analysis (Table S1 displays the analytes of
interest and the spectral region used for the fit). In some instances, peaks for the gases of interest
were also saturated in which case the pressure in the gas cell was reduced and the measurement
repeated.

**2.4 Calculation of emission ratios and emission factors**
A convenient quantity to compare emissions between burns is the emission ratio (ER). This ratio
is calculated by the change in the analyte of interest relative to the change in some known gas,
typically CO or $CO_2$.  For the present study, the change in analyte is divided by the change in CO:

$$\text{ER} = \left(\frac{\Delta\text{analyte}}{\Delta\text{CO}}\right). \tag{2}$$

It is important to note that these are the changes in analyte and CO relative to background
atmosphere (i.e. relative to ambient "clean air" conditions). The background levels of CO and $CO_2$
were measured using an open path gas analyzer (OPAG 22) prior to the series of burns. The initial
$CO_2$ level was measured to be 409 ppm (this value agrees with the global averaged $CO_2$ for May
2018 of 408.7 ppm (Dlugokencky and Tans)) whereas the CO level was often below the OPAG
detection limit. Without an instrument to measure ambient CO with sufficient sensitivity we
arbitrarily chose 200 ppb for an estimated background level which is within the range for a typical
CO mixing ratio (Seinfeld and Pandis, 2012). Emission ratios can be calculated for a single point
in time during the fire or they can incorporate the full length of the fire. The present ERs were
calculated based on the contents of the individual canisters and represent discrete ERs.  Other
studies have obtained fire-integrated ERs, which integrate over the entire duration of the fire (Koss

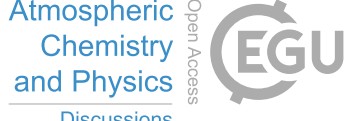

et al., 2018) or fire-averaged ERs determined from the slope of the regression with the intercept
set to zero (Yokelson et al., 1999).
Another useful quantity is the emission factor (EF), defined as the number of grams emitted of a
given analyte per kilogram of dry fuel consumed and estimated using the following equation
(Yokelson et al., 1999):

$$
\text{EF (g kg}^{-1}) = F_{carbon} \times 1000 \times \frac{MW_{analyte}}{MW_{carbon}} \times \frac{\frac{\Delta analyte}{\Delta CO_2}}{\sum_{j=1}^{n}\left(NC_j \times \frac{\Delta C_j}{\Delta CO_2}\right)}
$$


(3)


where $F_{carbon}$ is the mass fraction of carbon in the fuel, $MW_{analyte}$ and $MW_{carbon}$ are the molar masses
of the analyte and carbon, respectively, $\frac{\Delta analyte}{\Delta CO_2}$ is the emission ratio of the analyte relative to $CO_2$,
$\frac{\Delta C_j}{\Delta CO_2}$ is the emission ratio of species $j$ relative to $CO_2$ and $NC_j$ is the number of carbons in species
$j$. Note that $\Delta CO_2$ cancels out in equation 3. Elemental analysis of similar southeastern fuels was
reported in a previous study (Safdari et al., 2018), and the average carbon content by mass for
longleaf pine foliage and litter as well as sparkleberry was 0.52 which was used for $F_{carbon}$. (Table
S2 displays the elemental analysis for each fuel from Safdari et al., 2018).  One assumption in
equation 3 is that all of the carbon in the fuel is released and accounted for in the measurements of
the $j$ carbon species. Most carbon emissions are in the form of $CO_2$, CO or $CH_4$.  It should be noted
that the EF quantities reported here include only compounds measured by the FTIR, and EF values
may be overestimated by 1 to 2% due to undetected carbon species (Akagi et al., 2011).


**2.5 OH Reactvity**



Wildland fires release gases that may react with the hydroxyl radical (OH) to impact secondary
formation of ozone and downwind aerosols. To gauge the atmospheric chemistry effects of the
total gases emitted during the burns, the total OH reactivity (in units of $s^{-1}$ ppm $CO^{-1}$) for each of
the plots was determined by summing all of the ERs for each reactive gas multiplied by its
corresponding second order OH rate constant ($k_{OH}$ in units of $cm^3$ molecules$^{-1}$ $s^{-1}$) and a conversion
factor as outlined by Gilman et al. (2015). The conversion factor used for the present calculations
was $2.46 \times 10^{13}$ molecules $cm^{-3}$ ppm$^{-1}$. The rate constants were obtained from the NASA Panel
for Data Evaluation (Sander et al., 2006), Gilman et al. (2015), Atkinson et al. (2000) and the NIST
Chemical Kinetics Database.
**3. RESULTS AND DISCUSSION**

**3.1 Estimating the contribution from high and low temperature processes**
In a recent study Sekimoto et al. (2018) suggested that MCE may not be the best quantity to
adequately describe pyrolysis, but rather that emissions of volatile organic compounds (VOCs)
from biomass burning may be correlated with high and low temperature pyrolysis factors obtained
by carrying out positive matrix factorization (PMF) analysis on the emission profiles. The authors
further suggested that the ratio of acetylene ($C_2H_2$) to furan ($C_4H_4O$) could be used to estimate the
high and low temperature pyrolysis factors. They used the emission profiles from the analysis of
15 different fuels to calculate the following ratio that estimates the high and low temperature VOC
emissions:

$$\frac{(\text{Total VOC})_{\text{High T}}}{(\text{Total VOC})_{\text{Low T}}} = \frac{C_2H_2 \,/\, 0.0393}{C_4H_4O \,/\, 0.0159}.$$

(4)





We adopted this estimation approach and have used the acetylene to furan ratio to assess the
relative contributions from high and low temperature processes. The average results are displayed
in Figure 2 alongside the results from Koss et al. (2018), Gilman et al. (2015) and Akagi et al.
(2013). For comparison purposes, the values displayed in Figure 2 were determined using average
ERs for acetylene and furan.  The present results are approximately an order of magnitude greater
than all previous studies, likely due to the timing of collection and the sampling probe's proximity
to the flame. The juxtaposed values from the previous studies were obtained using either a) fire-
integrated ERs, b) discrete ERs sampled every 20 to 300 sec or c) fire-averaged ERs, all of which
incorporate several different phases of the fire as compared to the present measurements which
represent discrete samples just seconds before the flame front. With the Sekimoto et al. estimation
approach, higher acetylene/furan ratios indicate a greater contribution from the high temperature
process.  The markedly high ratio observed in this study suggests that samples were collected when
high temperature pyrolysis was indeed the dominant process. This observation is consistent with
the time profile for the contribution of the high temperature pyrolysis factor presented by Sekimoto
et al. (2018), which demonstrates that the contribution from high temperature pyrolysis [High-T /
(High-T + Low-T)] can easily exceed 0.95 in the early stages of fire, but reduces to smaller
fractions ($\leq 0.3$) in the latter stages. Another key difference is that the sampling probe used at Ft.
Jackson was positioned so as to extract gases directly before the flame front, yet in close proximity
to it, in order to limit further reactions. In particular, if the highly flammable acetylene molecules
were captured prior to subsequent oxidation reactions, this would explain the enhanced ratio of
high to low temperature VOC emissions as seen in Figure 2.






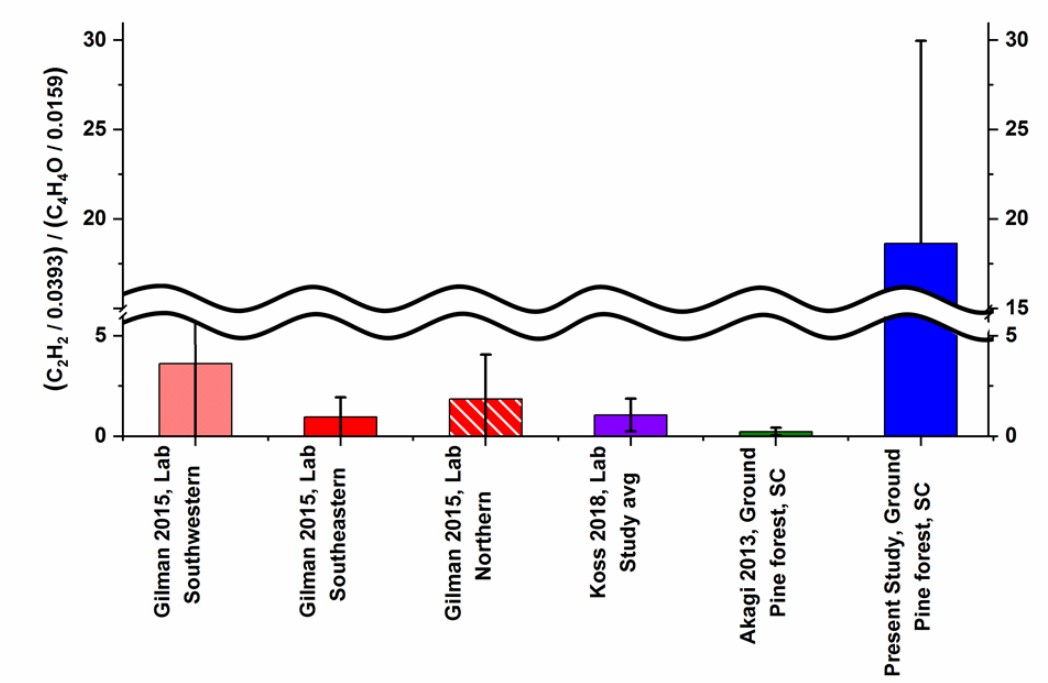


**Figure 2.** Ratio of acetylene ($C_2H_2$)/0.0393 to furan ($C_4H_4O$)/0.0159 to predict the ratio of high to low temperature
VOC emissions as outlined by Sekimoto et al. (2018). Error bars represent $1\sigma$. For the present study average results
were determined from the 10 collected samples preceding the flame front for acetylene and furan. Koss et al. (2018)
values were fire integrated while Gilman et al. (2015) used 20-300 sec integrations.  Akagi et al. (2013) reported EFs
which were used to calculate ERs for acetylene and furan.



## 3.2 CO$_2$ and CO Emissions and MCE Values

As expected, other than H$_2$O vapor, CO and CO$_2$ were the predominant gases observed as emissions. Table 2 displays the EF (g kg$^{-1}$) and ER (ppb/ppm$_{CO}$) values averaged for the ten field measurements, while Tables S3 and S4 contain the individual values for each measurement. The average MCE for all ten measurements was 0.83 ± 0.04, ranging from 0.75 to 0.87. Such MCE values would normally characterize data gathered during smoldering combustion where a combination of processes such as pyrolysis along with glowing combustion of char take place (Yokelson et al., 1997). Higher MCE values are associated with the flaming stage (~0.99 for pure flaming) and indicate more efficient combustion (i.e. a higher reaction temperature and more complete oxidation of the organic matter, while lower values (~0.65-0.85) are associated with the smoldering stage (Urbanski, 2013). Since the present study aimed at collection of pyrolysis gases preceding the flame front, characterizing the results in terms of MCE values may not be appropriate: The lower MCE values do not represent the fire being in the smoldering stage, but rather suggest that pyrolysis products were captured (at least in part) prior to the onset of combustion. As noted, the methodology used with this collection device ideally extracted the pyrolysis gases before they combusted. Due to the proximity of these gases to the flame and the surrounding atmosphere, however, air and combustion products in the region of sampling were likely captured along with pyrolysis gases.





**Table 2.** Study averages of MCE Study (0.83 ± 0.04), EF (g kg–1) and ER (ppb/ppm$_{CO}$) for
the ten measurements along with standard deviation (SD). SD represent the variation for the
ten non-identical measurements.

| Species | Formula | EF Study Average (g kg$^{-1}$) | SD | ER Study Average (ppb/ppm$_{CO}$) | SD |
|---|---|---|---|---|---|
| Carbon dioxide | $CO_2$ | 1469 | 113 | 5190 | 1450 |
| Carbon monoxide | CO | 191 | 45 | 1000 | n/a |
| Methane | $CH_4$ | 11.2 | 3.9 | 101.3 | 18.7 |
| Ethane | $C_2H_6$ | 1.14 | 0.42 | 5.54 | 1.48 |
| Ethene | $C_2H_4$ | 11.8 | 3.8 | 61.1 | 9.6 |
| Acetylene | $C_2H_2$ | 7.4 | 3.1 | 40.9 | 10.4 |
| Propene | $C_3H_6$ | 2.69 | 1.04 | 9.32 | 2.34 |
| Allene | $C_3H_4$ | 0.30 | 0.12 | 1.09 | 0.23 |
| 1,3-Butadiene | $C_4H_6$ | 1.20 | 0.72 | 3.13 | 1.25 |
| Isobutene | $C_4H_8$ | 0.23 | 0.15 | 0.58 | 0.31 |
| Isoprene | $C_5H_8$ | 0.63 | 0.90 | 1.18 | 1.43 |
| Naphthalene | $C_{10}H_8$ | 0.65 | 0.36 | 0.77 | 0.47 |
| Formaldehyde | HCHO | 0.76 | 0.98 | 3.63 | 4.57 |
| Methanol | $CH_3OH$ | 1.39 | 1.40 | 6.11 | 5.56 |
| Formic acid | HCOOH | 0.23 | 0.14 | 0.74 | 0.42 |
| Acetaldehyde | $CH_3CHO$ | 2.84 | 1.41 | 9.35 | 3.59 |
| Acetone | $(CH_3)_2CO$ | 1.15 | 0.77 | 2.92 | 1.78 |
| Acetic acid | $CH_3COOH$ | 1.45 | 2.66 | 3.46 | 6.15 |
| Acrolein | $C_3H_4O$ | 1.59 | 1.01 | 4.10 | 2.15 |
| Furan | $C_4H_4O$ | 0.41 | 0.25 | 0.89 | 0.49 |
| Furaldehyde | $C_4H_3OCHO$ | 1.01 | 1.01 | 1.45 | 1.31 |
| Hydrogen cyanide | HCN | 1.34 | 0.31 | 7.34 | 1.25 |
| Nitrous acid | HONO | 0.10 | 0.16 | 0.30 | 0.46 |
| Methyl nitrite | $CH_3ONO$ | 0.41 | 0.32 | 1.06 | 0.90 |




**3.3 Emissions of Lightweight Hydrocarbons**



Besides CO and $CO_2$, the second most abundant class of gases generated during the prescribed
burns was lightweight hydrocarbons (HCs). The lightweight HCs detected by the FTIR include
methane, ethane, ethene, acetylene, propene, allene, 1,3-butadiene, isoprene and isobutene. Most
of these (except allene) have been previously identified in fire emissions using FTIR either in
laboratory experiments (Burling et al., 2010; Christian et al., 2003; Christian et al., 2004; Gilman
et al., 2015; Goode et al., 1999; Hatch et al., 2017; Selimovic et al., 2018; Stockwell et al., 2014;
Yokelson et al., 1996; Yokelson et al., 1997) or field settings (Akagi et al., 2013; Akagi et al.,
2014; Alves et al., 2010; Burling et al., 2011; Goode et al., 2000; Hurst et al., 1994a; Hurst et al.,
1994b; Karl et al., 2007; Paton-Walsh et al., 2010). Figure 3 shows the individual correlations
between these lightweight HCs and excess CO mixing ratios. The analyte vs. ΔCO correlation
coefficients range from 0.97 (ethene and allene) to 0.66 (isoprene and isobutene). In all cases, the
correlation coefficients were larger with CO than with $CO_2$. Positive relationships have been
observed for CO correlations in previous burning studies (Hurst et al., 1994a; Hurst et al., 1994b).



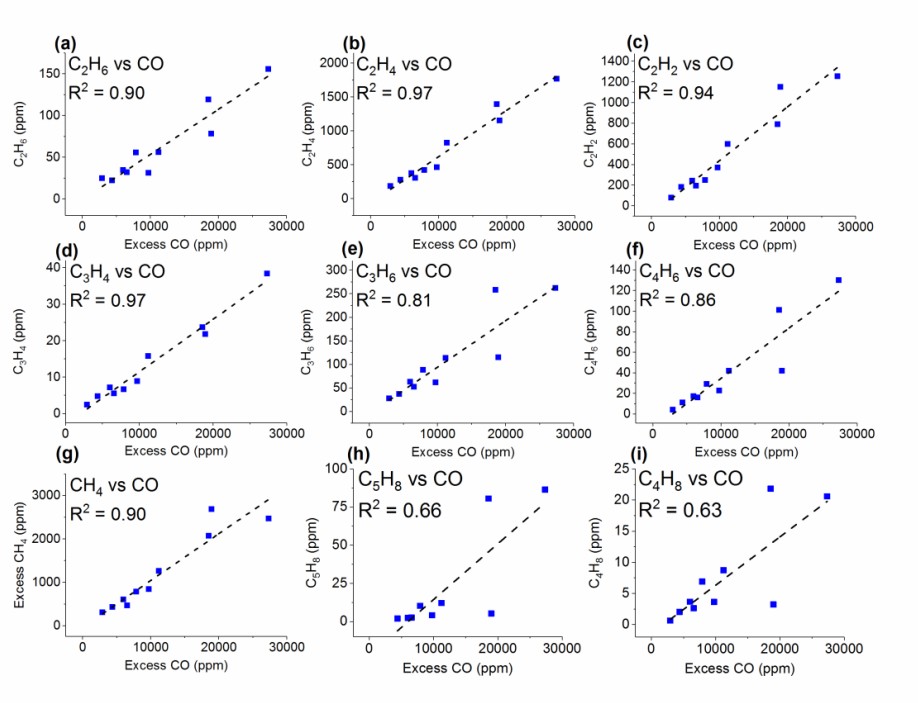


**Figure 3.** Mixing ratios (ppm) for the 10 measurements as a function of excess CO (ppm) for (a) ethane ($C_2H_6$), (b) ethene ($C_2H_4$), (c) acetylene ($C_2H_2$), (d) allene ($C_3H_4$), (e) propene ($C_3H_6$), (f) 1,3-butadiene ($C_4H_6$), (g) excess methane ($CH_4$), (h) isoprene ($C_5H_8$) and (i) isobutene ($C_4H_8$). The dashed lines are a linear fit to the data.


While the observed emission ratio (ER) for excess methane was comparable, ERs for ethene and
acetylene were considerably greater than previously reported values; specifically, Figure 4 shows
a comparison for methane, ethene and acetylene to previously reported values of Gilman et al.
(2015) and Akagi et al. (2013). As noted, different sampling methods complicate the comparison.
The present data represent a collection of instantaneous grab samples extracted directly before the
flame front, whereas the other data represent time averaged values. Ethene and acetylene have both
been observed as pyrolysis products in prior laboratory work (Palma, 2013), but may react further.
For example, the addition reaction of acetylene to benzene or naphthalene can produce styrene or
cyclopenta-fused polycyclic aromatic hydrocarbons (PAHs) (Ledesma et al., 2002). Alternatively





ethene and acetylene can undergo combustion (Simmie, 2003). Nevertheless, the high ER values
for ethene and especially for acetylene in the present study further suggest that the samples were
collected when the high temperature pyrolysis process was dominant; Sekimoto et al. (2018) also
observed that high temperature pyrolysis profiles are often associated with aliphatic unsaturated
hydrocarbons.

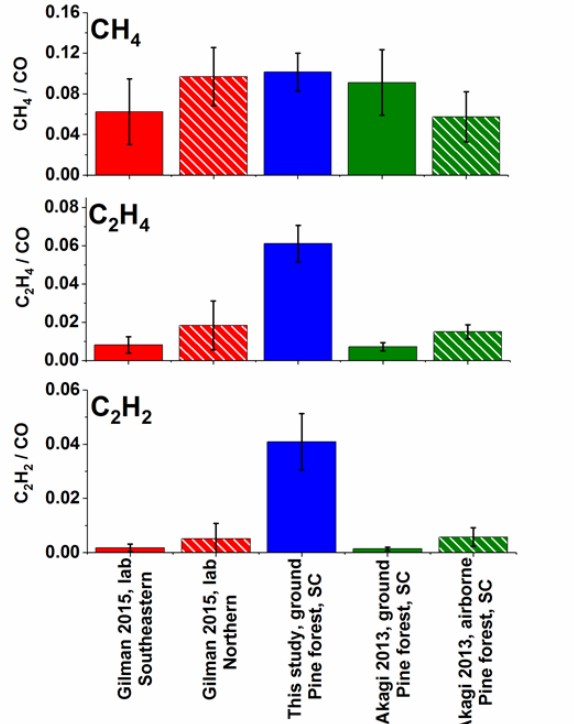

**Figure 4.**  Average emission ratios (ppm/ppm$_{CO}$) for excess methane (top), ethene (middle) and acetylene (bottom)
for this study and for previously published laboratory and field (ground and air based) investigations. Error bars
represent 1σ. Gilman et al. (2015) present discrete ERs with sample acquisition of 20 to 300 sec. Akagi et al. (2013)
present fire-averaged EFs calculated using ERs derived by the regression method. The emission ratios for Akagi et al.
shown above were derived from the ratio of the emission factors for the gas of interest and CO multiplied by the molar
mass of CO/molar mass of analyte.

**3.4 Emissions of Lightweight Oxygenated Hydrocarbons**
The noncyclic oxygenated hydrocarbons detected via FTIR analysis include formaldehyde,
methanol, formic acid, acetaldehyde, acetone, acetic acid and acrolein. Figure 5 shows average





ERs for these species along with the cyclic compounds furan and furaldehyde. On average,
acetaldehyde and methanol had the highest ER values in this group, with ER relative to CO of
0.009 and 0.006, respectively. Individual ERs per burn plot are displayed in Figure S2. Figure S2a
presents the results from site 16, while Figure S2b contains results from sites 24A and 24B. For
all measurements collected at sites 16 and 24A, acetaldehyde was consistently the highest with ER
values ranging from 0.005 to 0.014. Site 24B followed a different trend with highest ER values for
acetic acid, methanol, acetaldehyde and formaldehyde (in decreasing order). The ERs for acetic
acid and formaldehyde at site 24B are at least 7.9 and 2.5 times greater, respectively, than the other
burn sites: One key difference for site 24B was fuel composition, namely the presence and partial
consumption of larger logs (i.e. 7.6–20.3 cm diameter woody material). Other differences include
the presence of live pine seedlings and fewer turkey oak as compared to other plots. This particular
plot had the highest herbaceous and forb pre-fire loading and consumption with a higher fuel
moisture content (205% as compared to next highest value of 144%). This high fuel moisture
content was reflected in the ER for water, which was at least 4.7 times greater than the other plots
(Table S3). The pyrolysis of cellulose (one of the three primary components of biomass as
discussed below) forms levoglucosan. Shen et al. (2009) outline secondary decomposition
pathways for levoglucosan, in which the initial step is the rehydration to generate glucopyranose.
They demonstrate how glucopyranose can then form formaldehyde, methanol and acetic acid via
secondary decomposition routes. This pathway (or a similar one) may have been favored at site
24B. The greater ERs for acetic acid and formaldehyde observed at 24B may have been influenced
by the greater fraction of woody material and presence of herbaceous and forb fuels all with higher
moisture contents. This hypothesis warrants further investigation.



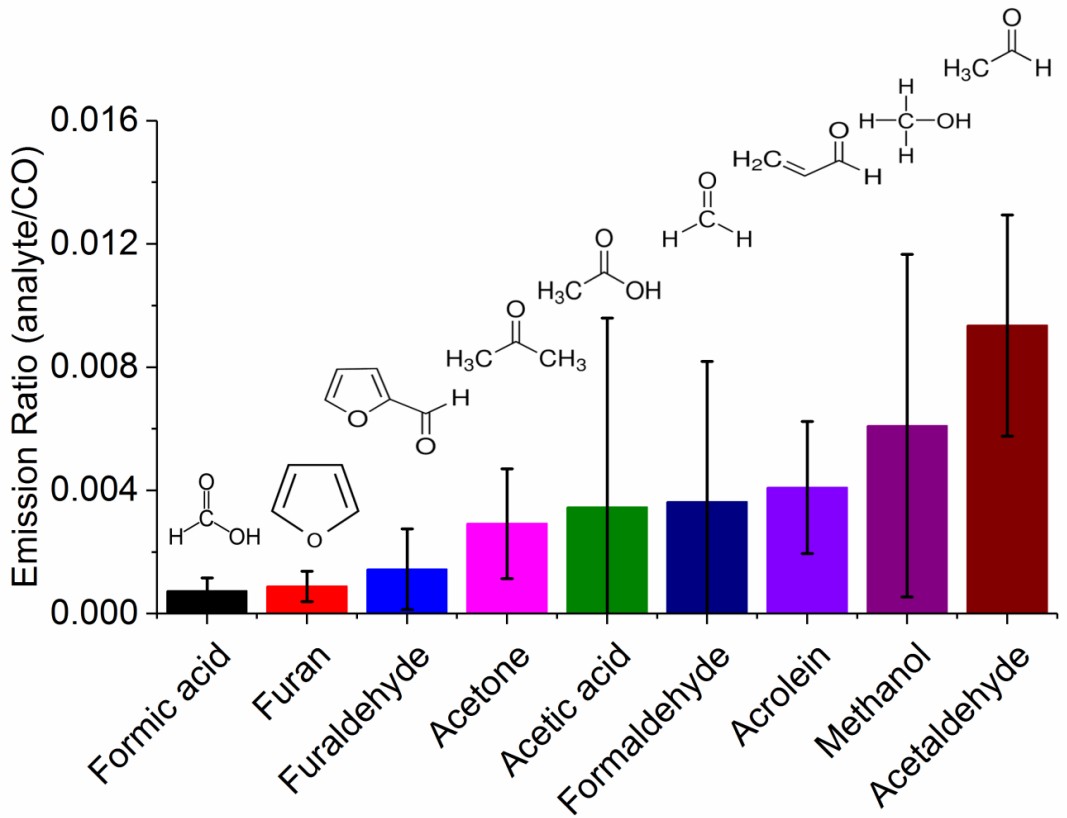

**Figure 5.** Average emission ratios (ppm/ppm$_{CO}$) for oxygenated hydrocarbons measured by FTIR for the 10 burn samples. Error bars represent 1σ.

Table 3 compares the present ER values with values from Akagi et al. (2013), Stockwell et al.

(2014), Gilman et al. (2015) and Koss et al. (2018). The present ERs are comparable to other burn

studies except for acetaldehyde, which appears to be marginally greater, and formaldehyde and

acetic acid, which both appear to be lower. The higher ratio for acetaldehyde may be due to

differences in the sampling approach as discussed above. That is, gas samples collected in the

present study may contain species that were generated during an earlier period in the thermal

decomposition process. In a controlled laboratory study by Stein et al (1983), acetaldehyde was

observed as one of the initial products emitted from the pyrolysis of glycerol, a product pyrolyzed

from levoglucosan. This same study also observed that acetaldehyde would continue to decompose



(under pyrolysis conditions) to smaller molecules such as ethene, methane, $H_2$ and CO (Stein et
al., 1983). The greater average ER for acetaldehyde observed in the present study may be due to
gases being captured (via the collection device) and removed from heat either in-between
decomposition steps or before combustion.

**Table 3.** Average emission ratios ($ppb/ppm_{CO}$) for this study and for previously published fire studies.

| Analyte | This Study- Pine forest SC ground- based | Gilman et al., 2015 southeastern Fuels | Koss et al., 2018 study average for all fuels | Stockwell et al., 2014 Sawgrass SC | Stockwell et al., 2014 Ponderosa Pine MT | Akagi et al., 2013 Pine forest SC ground- based | Akagi et al., 2013 Pine forest SC air-based |
|---|---|---|---|---|---|---|---|
| Formic acid | 0.7 | 1.6 | 2.2 | 0.7 | 5.1 | n/a | 0.6 |
| Furan | 0.9 | 0.7 | 1.9 | 0.8 | 1.2 | 2.4 | 1.1 |
| Furaldehyde | 1.5 | 1.5 | 2.1 | n/a | n/a | 0.1 | 0.2 |
| Acetone | 2.9 | 1.6 | 2.3 | n/a | n/a | 3.8 | 3.6 |
| Formaldehyde | 3.6 | 12 | 20 | 7.8 | 29 | 12 | 23 |
| Acetic Acid | 3.5 | 13 | n/a | 5.2 | 22 | 6.6 | 11 |
| Acrolein | 4.1 | 1.3 | 5.4 | n/a | n/a | 1.2 | 1.8 |
| Methanol | 6.1 | 7.8 | 12 | 3.4 | 24 | 21 | 13 |
| Acetaldehyde | 9.3 | 2.8 | 7.4 | n/a | n/a | 5.1 | 4.8 |

Koss et al. (2018) present the fire-integrated ERs. Gilman et al. (2015) present discrete ERs with sample acquisition of 20 to 300 sec. Stockwell et al. (2014) present the fire-integrated ERs. Akagi et al. (2013) present fire-averaged EFs calculated using ERs derived by the regression method. The emission ratios for Akagi et al. (2013) were obtained from the ratio of the emission factors for the analyte and CO multiplied by the molar mass of CO/molar mass of the analyte.


The slightly lower ERs for formaldehyde and acetic acid may in part be explained by secondary
decomposition pathways. Proposed pathways that generate formaldehyde and acetic acid proceed
through intermediates formed by the decomposition of levoglucosan (Shen and Gu, 2009).
Formaldehyde is generated from a number of intermediates such as hydroxyacetone (acetol)
(Lindenmaier et al., 2016) and 5-hydroxymethyl-furfural. While the formation mechanism for





acetic acid is via the decomposition of the intermediate hydroxyacetaldehyde (glycolaldehyde)
(Johnson et al., 2013), which undergoes a dehydration reaction to a ketene, and then a rehydration
to acetic acid (Shen and Gu, 2009), it is possible that the present conditions and fuels (save for site
24B) were not favorable for the above chemical pathways.
**3.5 Emissions of Aromatic Compounds**
In the present study, furan, furaldehyde and naphthalene were all detected via FTIR.  Previous fire
studies have used FTIR to detect phenol and/or furan (Burling et al., 2011; Akagi et al., 2014;
Hatch et al., 2017; Christian et al., 2003; Christian et al., 2004; Stockwell et al., 2014; Karl et al.,
2007; Selimovic et al., 2018; Yokelson et al., 2013; Burling et al., 2010; Akagi et al., 2013). One
of these studies also detected furaldehyde (Selimovic et al., 2018). To the best of our knowledge,
however, this is the first burning study that has used IR spectroscopy to identify naphthalene vapor,
though it has previously been detected in biomass burning emissions via other methods (Koss et
al., 2018; Gilman et al., 2015). Naphthalene has also been detected in tar samples generated from
the controlled pyrolysis of similar fuels (Safdari et al., 2018).

Phenol and phenolic compounds were not definitively observed in this study due to their IR bands
being somewhat weak and obscured by a number of other species, namely acetic acid, carbon
dioxide, acetylene and hydrogen cyanide. However, phenolic compounds have been identified in
products generated from the pyrolysis of lignin in controlled laboratory experiments by Kibet et
al. (2012). Lignin, one of the three main components of biomass, can account for 10–35% of the
biomass, and its chemical structure consists of polymers of various phenolic alkyl side chain
subunits (Shen et al., 2015). When undergoing thermal decomposition, lignin will release volatiles
at temperatures between 200 and 400°C. The proposed mechanism can generate intermediates



such as phenoxy radicals that ultimately lead to the formation of phenols (Kibet et al., 2012). In
the present study, spectral evidence of phenol was in fact observed in some measurements, but the
IR bands at 1176 and 752 cm$^{-1}$ were weak and were masked by other compound signatures,
hindering spectral quantification. Mixing ratios of phenol above the detection limit might have
been anticipated since prior controlled pyrolysis investigations of sparkleberry and longleaf pine
have detected phenol as a component in the tar (Safdari et al., 2018; Amini et al., 2019; Safdari et
al., 2019). While the phenol signal was weak, furan and furaldehyde, however, were clearly
detected, and their formation likely stemmed from thermal degradation of the other main
constituents of biomass.   Besides lignin, the other primary macromolecular components are
cellulose and hemicellulose, which account for approximately 50% and 15–35% by weight,
respectively (Shen et al., 2015). The pyrolysis of cellulose is known to produce furaldehyde, furan
and other low weight oxygenated compounds (e.g. acetic acid) via the intermediate levoglucosan
(Bai et al., 2013).   Moreover, furaldehyde and methanol have both been observed as volatile
products from the pyrolysis of methyl β-D-xylopyranoside, a model compound for xylan-based
hemicellulose (Shafizadeh et al., 1972).

Naphthalene is a polycyclic aromatic hydrocarbon (PAH) with several sources including a biomass
burning emission product. It was detected using FTIR for the first time in these studies (Scharko
et al., 2018). Its IR detection was not unexpected given that it has been observed in collected tar
samples generated by the laboratory pyrolysis of similar fuel types (Safdari et al., 2018) but its
identification in an experimental IR spectrum can be challenging as exemplified by Figure 6.  Most
of its IR bands have only moderate cross-sections with the exception of the $\nu_{46}$ band, which has a
strong Q-branch at 782.3 cm$^{-1}$ (green trace in Figure 6). For this band to be observed, however, it



needs to be deconvoluted from the acetylene rotational-vibrational lines also present in this spectral
domain (red trace in Figure 6). Better retrievals for naphthalene were obtained using a higher
spectral resolution (0.6 cm$^{-1}$) since the Q-branch of the $\nu_{46}$ band is quite sharp (FWHM ~ 1 cm$^{-1}$),
even at atmospheric pressure (Scharko et al., 2018).

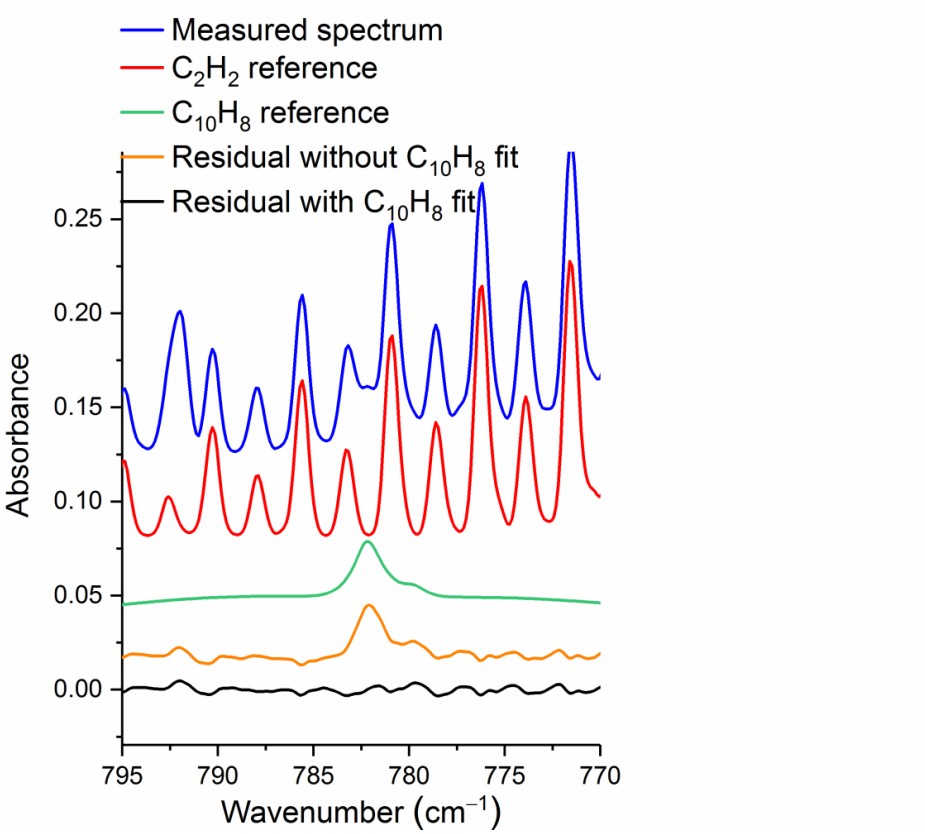


**Figure 6.** Measured and scaled reference spectra for acetylene ($C_2H_2$) and naphthalene ($C_{10}H_8$) as well as residual
with and without $C_{10}H_8$. The measurement is from site 16 plot 6 msmt. 2, and the detected mixing ratio for naphthalene
is 7.37 ppm. Spectra are offset for clarity. Reference absorption lines for acetylene are from HITRAN, and the
reference spectrum for naphthalene is from PNNL.

Figure 7a plots the mixing ratios (ppm) for naphthalene as a function of excess CO (ppm) while
Figure 7b displays the ERs for naphthalene for this study and previous studies.



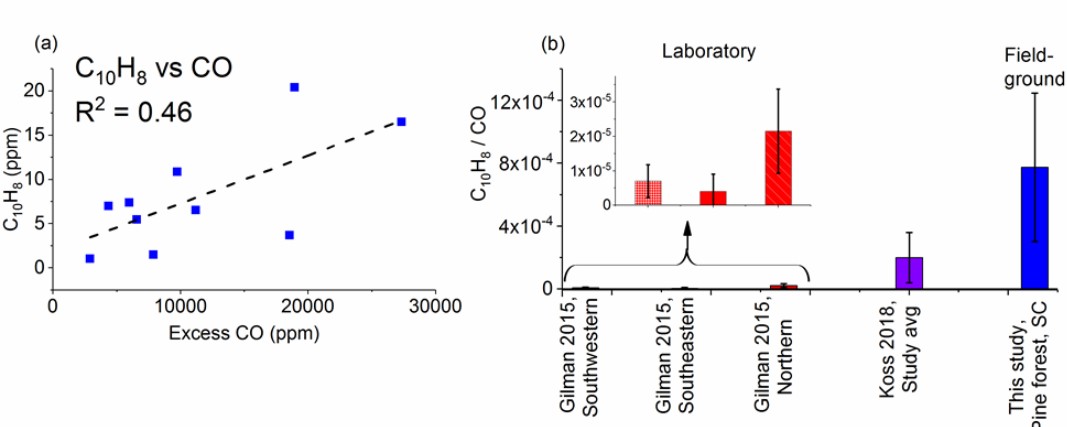

**Figure 7.** (a) Mixing ratios (ppm) for naphthalene ($C_{10}H_8$) as a function of excess CO (ppm) measured by FTIR for
each of the 10 canisters. The dashed line is a linear fit. (b) Average emission ratios ($ppm_{C_{10}H_8}/ppm_{CO}$) for this study
and for previous laboratory studies. Error bars represent $1\sigma$. Koss et al. (2018) present the fire-integrated ERs. Gilman
et al. (2015) present discrete ERs with sample acquisition of 20 to 300 sec.

The average naphthalene ER for this study is substantially greater than both the values from
Gilman et al. (2015) and Koss et al. (2018). The average for Koss et al. (2018), however, is in turn
an order of magnitude greater than the highest average for Gilman et al. (2015). The higher ER
for naphthalene in this study (shown in Figure 7) clearly suggests that the method to capture
pyrolysis gases was (at least in part) successful i.e. we were able to collect naphthalene gas prior
to it having undergone secondary reactions. Besides oxidation, under the right conditions
naphthalene can also continue to react forming still larger polyaromatics (Fairburn et al., 1990;
Richter and Howard, 2000). Sekimoto et al., (2018) also linked naphthalene with the high
temperature profile, and it appears that the samples in the present study were indeed collected
when the high temperature process was dominant. The detection of naphthalene suggests that
benzene and/or styrene, which are the main precursors to PAHs, may also be present. Styrene was
not detected via FTIR methods, and benzene is challenging for IR analysis since its one strong
band ($v_{11}$ mode at 673 cm$^{-1}$) is obfuscated by the $CO_2$ $v_2$ bending mode under polluted atmospheric





conditions. Figure S3 shows the 50°C PNNL reference spectrum of benzene in the spectral regions
where benzene's features are the strongest and experimental spectra (a) that are saturated and
unusable in that region and (b) that are not saturated and used for identification.

## 3.6 Emissions of Nitrogen-containing Species

Gases such as $NH_3$, $NO_2$, NO, HCN and HONO have been identified using FTIR spectroscopy in
fire laboratory experiments multiple times (Selimovic et al., 2018; Gilman et al., 2015; Christian
et al., 2003; Christian et al., 2004; Goode et al., 1999; Yokelson et al., 1996; Yokelson et al., 1997;
Stockwell et al., 2014; Hatch et al., 2017; Burling et al., 2010; Karl et al., 2007) as well as in field
studies (Yokelson et al., 1999; Burling et al., 2011; Goode et al., 2000; Akagi et al., 2013; Karl et
al., 2007; Akagi et al., 2014).  Multiple other methods have also been used to detect N-containing
gases, such as HNCO and $CH_3CN$ (Gilman et al., 2015; Christian et al., 2003; Christian et al.,
2004; Yokelson et al., 2009; Akagi et al., 2013; Karl et al., 2007; Roberts et al., 2010). The amount
and speciation of N-containing compounds emitted is dependent on fuel type and nitrogen content
(Stockwell et al., 2014; Burling et al., 2010; Coggon et al., 2016).  Moreover, emissions can usually
be linked to a stage of combustion: NO, $NO_2$, HNCO and HONO are all associated with the
flaming stage, while $NH_3$ and HCN are primarily associated with smoldering combustion but have
also been suggested as pyrolysis gases (Goode et al., 1999; Yokelson et al., 1996; Roberts et al.,
2010; Burling et al., 2010; Hansson et al., 2004; Di Blasi, 2008). Biomass pyrolysis experiments
carried out in an inert (i.e. oxygen free) atmosphere have revealed that $NH_3$, HCN and HNCO are
all generated (Hansson et al., 2004). These compounds are all considered to be $NO_x$ (NO + $NO_2$)
and $N_2O$ precursors because they are oxidized via combustion (Hansson et al., 2004).





The major N-containing compound identified in the present pyrolysis study was HCN. This is
consistent with previous small-scale and controlled laboratory studies that have shown HCN as
the primary N-product resulting from the pyrolysis of amino acids (Haidar et al., 1981; Johnson
and Kan, 1971).  This observation is further evidence that the gas samples were extracted when
high temperature was the dominant process; Sekimoto et al., (2018) have associated HCN with the
high temperature pyrolysis profile. Figure 8a shows the correlation between HCN and excess CO
($R^2$ = 0.89).  Previous field fire studies have observed similar trends (Simpson et al., 2011;
Stockwell et al., 2016). Figure 8b shows a comparison between the ERs for HCN for this study as
well as from previous laboratory and field (both ground and airborne) studies. The present values
are comparable to other ground-based measurements (Guérette et al., 2018; Akagi et al., 2013) but
differ from a few of the laboratory and airborne-based studies. It should be noted that although
conducted at a different time of the year (late Oct./early Nov. 2011), the studies by Akagi et al.
(2013) took place near the same location as the current study (i.e. the same military base), and the
ERs for HCN between the studies are not significantly different for the ground-based
measurements. This suggests that the ratio of initial gases released of HCN to CO is consistent
with the ratio of these gases over the duration of the fire, or at least the fire-averaged ratio. With
regards to ERs for HCN, the major factor that appears to influence these values is fuel type,
particularly the fuel's peat content. Both laboratory (Stockwell et al., 2014) and field (Stockwell
et al., 2016) studies of Indonesian peat have shown greatly enhanced ERs for HCN compared to
the studies represented in Figure 8b, which consist mostly of pine, grasses and fuels of non-peat
origin. The range in the averages of ERs for HCN shown in Figure 8b is 0.0028–0.0095; the
averages for the Indonesian peat in laboratory and field studies were 0.015 and 0.021, respectively



(Stockwell et al., 2014; Stockwell et al., 2016), and are considerably higher than the range of values
seen in Figure 8b.

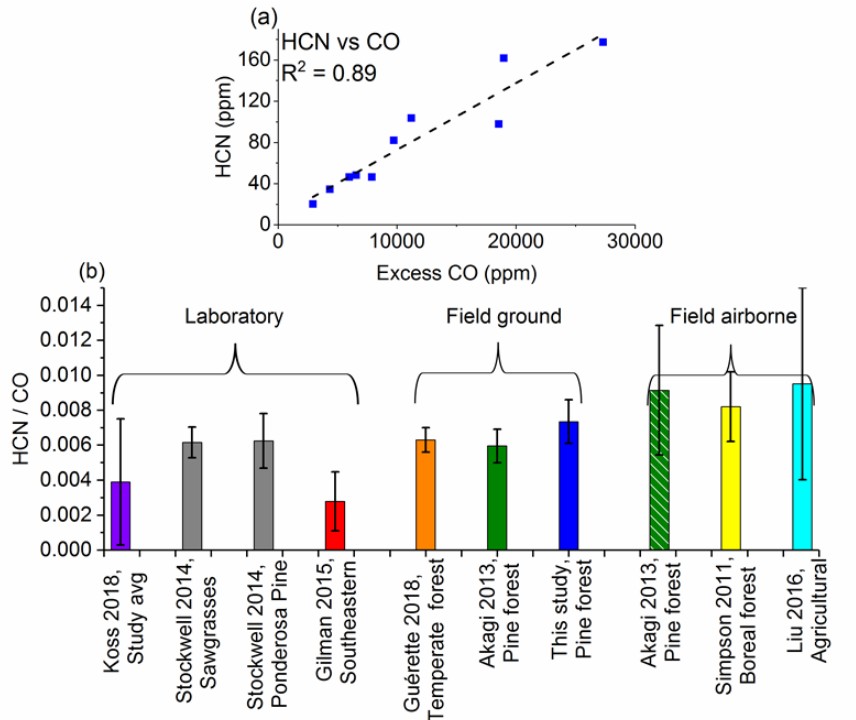

**Figure 8.** (a) Mixing ratios (ppm) for HCN as a function of excess CO (ppm) measured by FTIR. The dashed line is
a linear fit. (b) Average emission ratios (ppm/ppm CO) for this study and previous laboratory and field investigations.
Error bars represent 1σ. Koss et al. (2018) and Stockwell et al. (2014) present fire-integrated ERs. Gilman et al. (2015)
present discrete ERs with sample acquisition of 20–300 s. Simpson et al. (2011) present fire-averaged ERs derived by
regression. Guérette et al. (2018) present a single ER from all fires and derived by regression. Akagi et al. (2013) and
Liu et al. (2016) present fire-averaged EFs calculated using ERs derived by regression. The ERs for Akagi et al. (2013)
and Liu et al (2016) were derived from the ratio of the EFs for HCN and CO multiplied by the molar mass CO/molar
mass HCN.

In the present study, trace amounts of HONO were detected, but $NH_3$ was not observed. The
absence of $NH_3$ was somewhat unexpected since, similar to HCN, it is a known product from the
pyrolysis of amino acids (Haidar et al., 1981) and has been observed in prior prescribed fires
conducted at Ft. Jackson (Akagi et al., 2014; Akagi et al., 2013).   There are several possible
explanations for the lack of $NH_3$ in the measurements. First, Sekimoto et al., (2018) observed that



NH$_3$ is associated more with a low temperature pyrolysis profile, and it appears that the present
samples were extracted during a period when high temperature pyrolysis was the main process.
Second, NH$_3$ is strongly linked with the smoldering phase (Goode et al., 1999; Yokelson et al.,
1996), and samples were not collected during this phase. Third, the speciation of the N-species
emitted is dependent on the fuel composition and amount of oxygen (Ren and Zhao, 2013a, b,
2012), so it is possible that in the present study, the conditions favored HCN instead of NH$_3$.
Fourth, experimentally NH$_3$ is known to adhere to certain surfaces (e.g. steel), and in this study it
may have adhered to the canisters or tubing walls and was thus not detected.

The IR quantification of other N-species, such as NO, NO$_2$, CH$_3$NO$_2$ and HNCO was obstructed
due to interferences from H$_2$O, CO and CO$_2$ as well as the low emission values for some of these
N-species. Since NO and NO$_2$ are usually associated with flaming combustion, it was not
unexpected that these species were not observed. HNCO has been linked with pyrolysis processes,
and its main formation pathway is the cracking of cyclic amides along with HCN which is also a
product of pyrolysis, Hansson et al. (2004).

After accounting for the challenges in measuring NO, NO$_2$ and HNCO, the second most prevalent
N-containing species observed in this work was methyl nitrite (CH$_3$ONO). Methyl nitrite has
previously been detected in emissions from biomass burning using other methods (Gilman et al.,
2015). Figure 9a shows the plot of mixing ratios for methyl nitrite as a function of excess CO.
Unlike HCN (Figure 8a), methyl nitrite exhibits minimal correlation with excess CO. As one
possible alternative explanation, methyl nitrite is known to be associated with rocket-propelled





grenades (RPGs), but the Ft. Jackson military base records did not indicate RPG usage in the burn
plots (Scharko, 2019). While few fire studies have observed methyl nitrite, Gilman et al. (2015)
have detected it using GC-MS. Figure 9b shows a comparison of the results from Gilman et al.
(2015), separated by U.S. region, with the present results. It is worthy to note that both studies
observed similar ERs and that in the Gilman study, methyl nitrite had the second highest mean ER
after HCN for N-bearing species in southwestern fuels. Our observation of methyl nitrite is thus
not unprecedented, but this was its first reported detection via FTIR (Scharko et al., 2018). In the
present study, three measurements (Site 16, plot 1, msmt 1; Plot 24A, msmt 3; and Plot 24B) had
higher ERs for methyl nitrite than the others (Figure S4a displays the individual ERs for each
measurement), and it is unclear why this is the case. Other measurements collected at the same
location reported lower ER values. If the three highest ER measurements in question are not
included in the regression then the correlation between methyl nitrite and CO is stronger (Figure
S4b), and the average ER is closer to values reported by Gilman et al. (2015) for southeastern
fuels. One possible explanation for the three greater ER values is that the fuels may have contained
more components such as nitrate esters and isopropyl nitrate, both of which are known to release
minor amounts of methyl nitrite under controlled pyrolysis conditions (Boschan et al., 1955;
Griffiths et al., 1975).

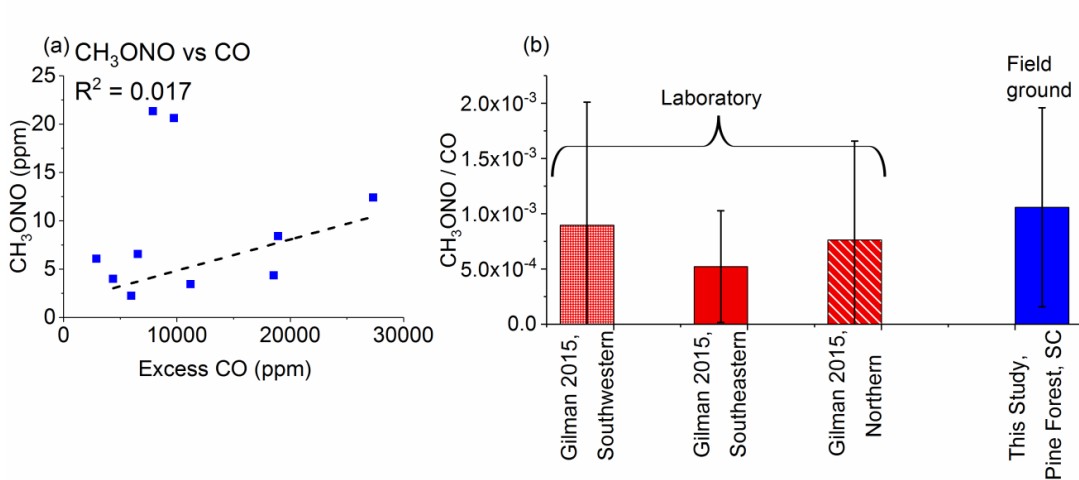

**Figure 9.** (a) Mixing ratios (ppm) for methyl nitrite ($CH_3ONO$) as a function of excess CO (ppm) as measured by
FTIR. The dashed line is a linear fit. (b) Average emission ratios (ppm/ppm CO) for this study and previously
published study carried out in the laboratory. Error bars represent 1σ. Gilman et al. (2015) present discrete ERs with
sample acquisition of 20 to 300 s.

**3.7 OH Reactivity**
Most gases released from biomass burning will undergo secondary chemistry in the atmosphere.
Other than photolysis or reactions with $NO_3$ or $O_3$, the primary destructive mechanism for most
compounds is their oxidation by the OH radical (i.e. it governs the lifetimes of most components
in the atmosphere). Along with the presence of VOCs and $NO_x$, the OH radical contributes to the
formation of ozone and particulate matter (aerosols) (Atkinson, 2000; Finlayson-Pitts and Pitts Jr,
1999). In view of that, ozone and secondary organic aerosols have been measured in previous field
biomass burning studies (Yokelson et al., 2009; Akagi et al., 2013). Additionally, prior fire studies
have used OH reactivity as a metric to identify reactive species that may impact downwind
chemistry (Gilman et al., 2015; Koss et al., 2018). OH reactivity has been defined as the loss
frequency of OH due to its reaction with reactive species in the atmosphere and generally is
expressed in units of $s^{-1}$ (Zannoni et al., 2016). Here the quantity is relative to CO, and the units
are $s^{-1}$ ppm $CO^{-1}$. By multiplying the emission ratio by the associated OH rate constant, the



resulting quantity can provide an indication of which species are most likely to lead to ozone and
aerosol formation as a plume ages.

The OH reactivity was determined by the method outlined in Section 2.5 for each compound using
the ERs (Table S3) and previously reported OH rate constants (Table S6). The compounds were
grouped into four categories depending on their chemical structures. Alkanes, nitrogen containing
species and naphthalene were subsequently grouped together for clarity because their values were
low and not visible in Figure 10. The average total OH reactivity was $39.4 \pm 9.8$ s$^{-1}$ ppm CO$^{-1}$,
and the percent contributions per category are displayed in Figure 10a. The category with the
greatest percent contribution was alkenes with 75%. Figure 10b displays percent contribution for
each alkene to the total alkene OH reactivity. The specific alkene that had the greatest OH
reactivity was ethene with an average of $12.8 \pm 2.0$ s$^{-1}$ ppm CO$^{-1}$, which was more than 1.8 times
the second highest value (propene with an average of $6.9 \pm 1.7$ s$^{-1}$ ppm CO$^{-1}$). In a prior fire study,
Gilman et al. (2015) observed the largest contributions from propene (study average of 3.5 s$^{-1}$ ppm
CO$^{-1}$ using the same $k_{OH}$ found in Table S6) and second largest with ethene (study average of 2.4
s$^{-1}$ ppm CO$^{-1}$ using the same $k_{OH}$ found in Table S6). Although propene has a faster OH rate
constant than ethene, the high ERs observed for ethene in this study consequently enhanced its OH
reactivity above that of propene.

As previously discussed, the higher ERs for ethene (and other compounds, e.g. acetylene) are
likely due to high temperature pyrolysis being the dominant process with some pyrolytic gases
escaping the flame front. If that is indeed the case, the OH reactivities presented here reflect

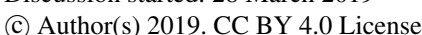



emissions that have partly escaped some secondary pyrolysis and combustion reactions. Higher
ERs for acetylene were also observed; however, due to its relatively slow reaction with OH, its
average reactivity was $0.78 \pm 0.2$ s$^{-1}$ ppm CO$^{-1}$ (~2% of the total). Ethene, however, reacts faster
with OH. This OH-initiated reaction generates a hydroxyethyl radical, which propagates radicals
in the atmosphere through a series of reactions, leading to formation of both formaldehyde and
glycolaldehyde (Orlando et al., 1998).

The second category with the greatest fractional contribution was the oxygenated HCs, and the
major contributors were acetaldehyde and acrolein with reactivities of $3.7 \pm 1.4$ and $2.0 \pm 1.4$ s$^{-1}$
ppm CO$^{-1}$, respectively. Both of these aldehydes react with OH to form CO and formaldehyde as
well as peroxyacetylnitrate (PAN) for acetaldehyde (D'Anna et al., 2003) and
acryloylperoxynitrate (APAN) for acrolein (Orlando and Tyndall, 2002). Higher ERs for
acetaldehyde were observed. However, acrolein had a comparable average to the laboratory study
by Koss et al. (2018). The other two categories (furan and alkanes, nitrogen containing species
and naphthalene) had OH average reactivities of $2.1 \pm 1.5$ and $0.53 \pm 0.3$ s$^{-1}$ ppm CO$^{-1}$,
respectively. The individual OH reactivities for each measurement are displayed in Figure S5.
Consistent with the average, the category with the greatest fractional contribution was alkenes
followed by oxygenated HCs. Nonetheless, the measurement that had a significantly greater
contribution from the oxygenated HCs group (and less from alkenes) was site 24B. For site 24B,
the percent contribution of alkenes was 49% (study average = 75%) and for oxygenated HCs was
38% (study average = 18%).



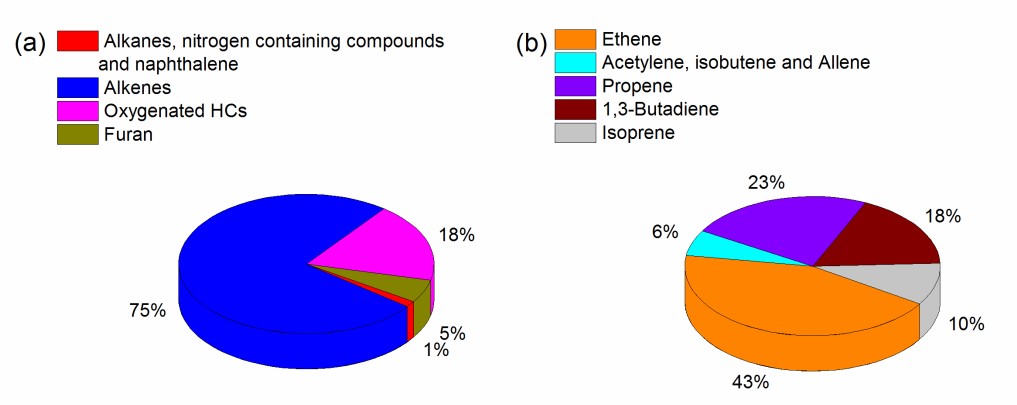


**Figure 10.** (a) Average percent contributions for the total OH reactivity for all ten measurements ($39.4 \pm 9.8$ s$^{-1}$ ppm
CO$^{-1}$) and (b) the average percent contributions for each alkene to the total alkene OH reactivity ($29.5 \pm 8.7$ s$^{-1}$ ppm
CO$^{-1}$). Error represents $1\sigma$.


## 4. CONCLUSIONS


The objective of this study was to collect and quantify gas-phase compounds emitted ahead of the
flame front (prior to the onset of combustion) in prescribed burns conducted in a pine forest.
Primary and secondary decomposition pathways generate volatile products, which act as fuel gases
that can undergo combustion and contribute to sustaining the fire. The main observations are that
the estimated ratio of high to low temperature VOC emissions suggest that the samples were indeed
extracted when the high temperature pyrolysis process was dominant. The acetylene/furan ratio
suggested by Sekimoto (2018) was nearly 10x higher than previous studies; this is in fact consistent
as previous works all had longer collection times, in some cases fire-averaged values. The
significantly greater ERs observed for specific compounds, e.g. lightweight HCs such as ethene
and acetylene as well as unoxidized aromatics such as naphthalene all support the hypothesis that
the grab samples were collected prior to onset of decomposition, recombination or combustion
reactions, and that such gases represent pyrolytic processes. For the oxidized organics,



acetaldehyde and methanol consistently had the highest ER values relative to CO for this collection

of pyrolysis gases. The ERs for acetic acid and formaldehyde were found to be high in some

instances, but this appeared to be related to fuel composition of the individual burn site.  The major

N-component released was HCN, while $NH_3$ was not observed, which is consistent with the

collected gases representing species associated with the high temperature pyrolysis process.

## ASSOCIATED CONTENT

### Author contribution

NKS, TLM, and TJJ contributed to the writing of this manuscript. AMO and RGT set up laboratory

and recorded infrared data. NKS, AMO and CAB provided data processing and analysis. SPB,

ENL, JC, BMC and GMB aided in collection of field samples. JC provided thermal imaging and

videography. RDO and JRC contributed to fuel characterization. DRW and TJJ were the project

managers.

### Supporting Information

The Supporting Information is available free of charge.

### ACKNOWLEDGMENT

This work was supported by the Department of Defense's Strategic Environmental Research and

Development Program (SERDP), Project RC-2640 and we gratefully acknowledge our sponsor for

their support. PNNL is operated for the U.S. Department of Energy by the Battelle Memorial

Institute under contract DE-AC06-76RLO 1830. We gratefully thank David W. T. Griffith for his

valuable guidance and direction using the program MALT5 for spectral analysis. We are grateful

to John Maitland and colleagues at Fort Jackson for hosting the field campaign and carrying out





696    the burns. We thank Olivia Williams for help with spectral analysis using MALT5. In addition, we

697    are thankful to Professor Michael L. Myrick and his students at the University of South Carolina

698    for hosting us in their laboratory and for their helpful support setting up the instrument.



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
