# Peer review of "Gas-Phase Pyrolysis Products Emitted by Prescribed Fires in Pine Forests with a 1 Shrub Understory in the Southeastern United States\*\* 2 3 4 Nicole K. Scharko1, Ashley M. Oeck1, Tanya L. Myers1, Russell G. Tonkyn1, Catherine A. Banach1, S"

_Atmospheric Chemistry and Physics, 2019_

## Referee Comment (RC1) · Anonymous Referee #1 · 25 Apr 2019

Review of Gas-Phase Pyrolysis Products Emitted by Prescribed Fires in Pine Forests with a Shrub Understory in the Southeastern United States ACP 2019-174 Scharko et al.

Summary:

The authors collected canister samples of traces gases directly ahead of the flame front during a series of prescribed burns. Fuels were representative of the Southeastern US forest with some variability in fuel speciation and moisture content. Trace gas speciation and concentration were determined offline by FTIR. Emission factors and emission ratios were calculated and compared to previously reported values from the literature. VOCs were categorized by functional group and the FTIR analytical capability with each functional group is discussed. The VOC composition is compared to metric of high/low temp pyrolysis, and to MCE. Samples from this study were consistent with dominantly high-temperature pyrolysis. MCE did not appear to correctly categorize emissions. The authors find in general much higher emission ratios for many species than have been previously reported. The discrepancy may be due to targeted collection of VOCs from one fire process (high-temperature pyrolysis) as opposed to a broad sampling of many fire processes.

Major comments:

In general I find this a well-written paper with no major issues. The methodology is appropriate and the data are presented in a well-organised way. The analysis is straightforward. The paper adds to the body of work reporting VOC composition from fires in a non-laboratory setting, and should be especially useful to operators of FTIR. I have only a few comments and suggestions listed below.

Supplementary materials were referenced in the manuscript but are not accessible online.

The paper would benefit from a tighter definition of the specific fire process being targeted, especially the distinction between pyrolysis and the flaming stage.

I don't have a good sense of how important the pre-flame pyrolysis emissions are compared to total emissions from a fire. Do the authors have a metric of what fraction of VOCs (or OH reactivity) is emitted in this process?

Specific comments:

Line 220: What is the OPAG CO detection limit?

Section 3.1: Since many VOCs were measured during this study, a more robust method

of estimating high vs. low pyrolysis fraction could be to fit the entire VOC profiles in ppb provided in Sekimoto et al. 2018 to the measured species. It could also be helpful to compare the overall VOC profile measured here to the high-temperature pyrolysis profile.

Table 2: MCE is not reported in the table although it is in the table title.

Figure 3: The reported mixing ratios (parts-per-thousand) seem excessively high; have similarly high mixing ratios ever been reported previously in the outdoor environment?

Figure 5: I'm not sure this figure adds anything to the paper and it could be removed. What do the colours indicate?

Line 484: What is meant by secondary reactions? Photolysis or reaction with OH? Or further pyrolysis reaction? Gilman et al. 2015 and Koss et al. 2018 are laboratory studies, so atmospheric reaction seems unlikely. What reaction could reduce the level of naphthalene a factor of 10 on the timescale of a laboratory study? Or is the reduction in ER an effect of elevated CO in the other studies?

---

## Referee Comment (RC2) · Anonymous Referee #2 · 1 May 2019

General Comments:

This is interesting work attempting to identify and quantify emissions specifically from the "pyrolysis process" preceding combustion from a prescribed fire. Trace gases were sampled using canisters and quantified offline by FTIR. Emission ratios and emission factors were calculated and compared to a few previous publications. The authors use Sekimoto et al. 2018's ratio of ethyne to furan along with reporting higher ERs for several species compared to other studies as evidence that there was selective sampling of high-temperature pyrolysis emissions. Altogether, this is insufficient evidence that

pyrolysis-only emissions were selectively captured by the custom sampling device. It is unclear how this body of work is useful to modelers as pyrolysis products are only part of the total fire emissions necessary to accurately model fire. While it does little to inform us about best practices for prescribed burn conditions, it is an interesting attempt to isolate pyrolysis compounds in an "uncontrolled" real-world setting, however, there are entire journals (e.g. Journal of Analytical and Applied Pyrolysis, Fuel, etc.) dedicated to identifying the chemical products from pyrolysis. There is insufficient motivation for why this data is useful and the authors have not made a strong case for publication. It is likely these partial fire results have a strong potential to mislead or confuse the casual reader.

Specific comments in no particular order:

(1) In the introduction, a lot of emphasis is placed on the importance of identifying and quantifying biomass burning emissions for model predictions, which guide best practices for prescribed burning conditions. Additionally, the authors highlight the complexities of fires and emission dependencies on fuel types, burn techniques, geometry, etc., however, there isn't proper support/justification for why measurements of just the pyrolysis emissions is vital for these sorts of model predictions or for analyzing burn techniques. A more "realistic" set of emission factors is obtained following combustion as many of the products measured during "pyrolysis only" will likely be consumed and/or combusted in the flame at higher temperatures resulting in different chemical products.

(2) The introduction is quite long and needs to remain more focused. It covers everything including downwind formation measured by FTIR to a significant discussion of MCE. It may be useful to rearrange and focus the introduction on motivation and important background considerations of FTIR sampling and pyrolysis measurements and move MCE discussions later in the manuscript.

(3) P4 L 80 There is some discussion in the introduction about the importance of

prescribed burning and here the authors reference the necessity to understand burn conditions to minimize gases released during prescribed burns. It may be persuasive to highlight Liu et al., 2017, which provides clear evidence that the PM emitted from a wildfire is much greater than that for a controlled prescribed burn. While the authors have steered clear of PM discussion, this still highlights some major differences observed between the two. It may also be useful to highlight some of the differences between a prescribed burn versus a wildfire in terms of fuel consumption as well as considerations on meteorological and fuel moisture conditions. Liu, X., et al. (2017), Airborne measurements of western U.S. wildfire emissions: Comparison with prescribed burning and air quality implications, J. Geophys. Res. Atmos., 122, 6108–6129, doi:10.1002/2016JD026315

(4) P4 L 89 It is incorrectly stated that pyrolysis is the "first step" in the burning process. Initially there is evaporation of water and other gases absorbed to the solid fuel (distillation), then once the temperature gets high enough to break bonds in the solid fuel both organic aerosol and non-methane organic gases are given off (pyrolysis) and this flammable mixture serves as the airborne fuel that can be rapidly oxidized. Some of the pyrolysis products are processed in the flame to $CO_2$ and BC, however, much of it escapes unoxidized. Entrainments of unoxidized pyrolysis products in smoke naturally falls within definition of smoldering combustion (McKenzie et al., 1995). It might be useful to provide some detail and description of these various processes. It is also important to note that "smoldering" is a complex mixture of multiple processes. For instance, P16 L 308 the authors suggest they aren't measuring products from smoldering combustion, however, distillation, glowing, and pyrolysis are often grouped into smoldering combustion, therefore where is the evidence you are measuring pyrolysis only? McKenzie, L. M., W. M. Hao, G. N. Richards, and D. E. Ward, Measurement and modeling of air toxins from smoldering combustion of biomass, Environ. Sci. Technol., 29, 2047 – 2054, 1995.

(5) The goal was to sample before the flame front, though it is likely that glowing com-
bustion is also hot enough to drive pyrolysis. Additionally, the high heat release rate associated with flaming combustion likely pyrolyzes more fuels and there is often a peak in pyrolysis products before, during, or shortly after the peak in flaming product. Here the authors are likely probing only a portion of the pyrolysis products as sampling preceded the flame front only. There was no attempt to sample pyrolysis products escaping oxidation in the cool flame interiors or following the flame front. Thus, the "pyrolysis products" measured are likely biased/limited to products specifically sampled during the flame front sampling, and it has been shown in other work that pyrolysis mostly occurs after the flame front driven by the heat of glowing (Yokelson et al. 1996). For these reasons, how representative are the pyrolysis emissions measured here?

Yokelson, R. J., D. W. T. Griffith, and D. E. Ward, Open-path Fourier transform infrared studies of large-scale laboratory biomass fires, J. Geophys Res., 101, 21,067 – 21,080, 1996.

(6) It doesn't appear as though there is any temperature data available, thus how can the authors distinguish between "low-temperature" and "high temperature" pyrolysis? The assignment is purely qualitative. Figure 1 only shows temperatures up to 170C. We'd expect in an oxygen-rich environment the biomass is readily ignited around 500-600C, therefore most "high temperature" pyrolysis experiments are performed in a N2-atmosphere and more accurately report pyrolysis specific compounds. Determining and quantifying pyrolysis products across a range of temperatures and fuels would have been more useful by heating fuels in a laboratory where the FTIR could instead sample directly.

(7) There is some discussion about spectral regions used and MALT conditions, however, there is no discussion concerning the uncertainty and LODs of your measurements. While much of this may be discussed in Scharko et al. 2018, there still needs to be some information provided in this manuscript.

(8) The detection limits of 12 ppm for acetaldehyde described in the companion paper

on this technique are not sensitive enough for most real-world smoke. Do the high LODs of other compounds prevent their detection in this work?

(9) Similar to whole air sampling canister-based measurements, there needs to be some discussion about the losses of compounds to the walls of the canister, compounds reacting away, as well as gases formed. Could this account for the higher ERs of alkenes such as ethyne? There is a brief mention of wall losses of NH3, and I suspect this is the major reason no ammonia was measured in this study. The authors incorrectly state that the absence of NH3 supports the fact that low-temperature pyrolysis and/or smoldering isn't measured in this study. It would be useful to cite other FTIR measurements and discussions of ammonia wall losses as well as the potential wall losses of other "sticky compounds" (Stockwell et al., 2014; Yokelson et al., 2003). Yokelson, R. J., Christian, T. J., Bertschi, I. T., and Hao, W. M.: Evaluation of adsorption effects 10 on measurements of ammonia, acetic acid, and methanol, J. Geophys. Res., 108, 4649, doi:10.1029/2003JD003549, 2003.

(10) How useful is calculating an emission factor, considering the definition of EF is the amount of a compound emitted per unit of dry fuel consumed? If measurements only probe the beginning of the process before any combustion has occurred, then the fuel has not been completely consumed. Does this complicate the use of EFs and the assumptions about the amount of carbon consumed considered in the emission factor calculation?

(11) P13 L 249 – The authors mention the atmospheric chemistry effects of "total gases emitted during the burns" being the motivation for calculating OH reactivities, however, this is not really an accurate statement as this manuscript is only measuring pyrolysis products preceding combustion. More realistic EFs would be calculated from fire-integrated EFs. Is there other motivation for including a section in the manuscript on OH reactivity?

(12) Eqn 4: Sekimoto et al., 2018 hasn't shown how effective the use of these two

proxies are for real-world fires and the equation was designed to predict total VOC emissions. Using this equation to justify sampling of "high temperature" pyrolysis is not supported. Additionally, in Figure 2 how have the authors decided to compare to these particular studies? Would it be more useful to compare to other field studies, instead of using so many laboratory fire-integrated emissions? It may be useful to focus on field studies with similar fuel types or similar burn conditions (e.g. Yokelson et al 2013, Akagi et al., 2013 "late EFs" calculated for smoldering combustions, Bertschi et al., 2003 samples residual smoldering combustion).

(13) How did the authors ensure they weren't getting flaming products from fire-driven flow? $C_2H_2$ has been observed to increase with increasing MCE (Burling et al., 2011; Yokelson et al 2008) and it's been noted as a "flaming" compound previously. I understand it can also be produced from smoldering combustion, but what makes the author's certain they are capturing pyrolysis produced $C_2H_2$?

(14) It is unclear how useful some of the tables and figures are. The comparisons to previous work seems arbitrary. Some justification for how studies were selected for comparisons are needed.

Technical corrections:

P2 L 39 Should be "biomass burning emissions"

P3 L 54 More up-to-date citations could be helpful here

P3 L 71 Similar to specific comment (2): There is some detailed discussion about the downwind changes in fire emissions measured by FTIR. It's not entirely clear why this is relevant in the discussion as no downwind smoke is measured in this study.

P4 L 84 & L 86 I don't think "hotter" and "cooler" is proper terminology. There is quite a range in temperatures in a fire, both in the flame itself and throughout the fuel bed, thus hotter/cooler are relative terms.

P4 L 86-87 It might be worth mentioning more OVOCs from smoldering combustion at

lower MCE

P6 L 126 "attempts to ensure" Please change this phrasing, it is confusing.

P10 L 182 I don't understand the "aliquots"- Does this mean samples in the same canister were all taken from different areas at different times?

P11 L 204 I'm confused about your $H_2O$ and $CO_2$, this wording implies that you did not actually measure these species, though I'm assuming that instead the authors used different spectral regions that weren't optically saturated as values are reported for these compounds later on?

P11 L 212- Instead of relative to any "known gas" it should be relative to a co-emitted, long-lived species. Some discussion to describe that CO and $CO_2$ can be generated during glowing, pyrolysis, and flaming would be useful.

P11 L 216- The background levels of both CO and $CO_2$ likely changed as fuels were burned in the area. I noticed several burns occurred in time fairly close together, but it is not clear whether the authors took into account the possibility of a changing background or if a constant background was assumed. I believe it is the latter, but this likely adds uncertainty to the ERs.

P11 L224- The authors state these are discrete ERs, however, it seems as though there are several samples in the same canister taken from various areas at various times, "aliquots" as they are described. Because of these sampling preferences, is this really a "discrete" sample and how comparable are these to other more traditional grab "snapshot" samples generally seen when sampling with canisters?

P12 L 230 also reference Ward and Radke, 1993 Ward, D. E., and L. F. Radke, Emissions measurements from vegetation fires: A comparative evaluation of methods and results, in Fire in the Environment: The Ecological, Atmospheric and Climatic Importance of Vegetation Fires, edited by P. J. Crutzen and J. G. Goldammer, pp. 53 – 76, John Wiley, New York, 1993

P12 L 243- Stockwell et al. 2015 found these underestimates can be higher for certain fuel types Stockwell, C. E., Veres, P. R., Williams, J., and Yokelson, R. J.: Characterization of biomass burning emissions from cooking fires, peat, crop residue, and other fuels with high-resolution proton-transfer-reaction time-of-flight mass spectrometry, Atmos. Chem. Phys., 15, 845–865, https://doi.org/10.5194/acp-15845-2015, 2015

P13 L 260 This first sentence is not entirely accurate. The combustion efficiency equation is designated to describe combustion chemistry and Sekimoto shows that MCE doesn't necessarily describe their VOC emission profiles. It is true that MCE does not correlate with products as well for pure smoldering, especially for grab samples or in real time, because the same MCE may characterize different white smoke/glowing ratios. MCE is most useful when there is a range of flaming/smoldering as is common in most real-world fires, however, this is not the case in this manuscript.

P16 L308- The authors state: "the lower MCE values do not represent the fire burning in the smoldering stage" This is an unsupported claim and as I mentioned earlier smoldering is a grab bag for various processes that include distillation, glowing, and pyrolysis.

P16 L 311 This may be a good area to mention fire-driven flow

P17 Figure 2: where are the study average MCEs?

P18 L 336- What was the range of $CO_2$ emissions? If only a narrow spread, this might explain the lower correlation coefficients for something like $C_2H_2$ with $CO_2$.

Table 3. Stockwell et al 2015 has a more complete list of compounds measured including acetaldehyde, acetone, etc. Stockwell, C. E., Veres, P. R., Williams, J., and Yokelson, R. J.: Characterization of biomass burning emissions from cooking fires, peat, crop residue, and other fuels with high-resolution proton-transfer-reaction time-of-flight mass spectrometry, Atmos. Chem. Phys., 15, 845-865, https://doi.org/10.5194/acp-15-845-2015, 2015.

P29 L 531 The fuel type and N-composition certainly influences HCN emissions, however, it is random to highlight Indonesian peat here, this is a very unique fuel type and peat burns by smoldering combustion only, thus it follows that the fuel N is biased towards HCN rather than flaming products such as NOx. Not sure it is worth mentioning in this manuscript.

P31 L 555 The authors argue NH3 is only present in the smoldering phase. I'd argue you did measure some smoldering combustion and the most likely for no NH3 detection is wall losses in the canisters as I discussed in specific comment (9).

---

## Author Comment (AC1) · 19 May 2019

Responses to Referee Comments for ACP 2019 Referee #1 Referee Comments: In general I find this a well-written paper with no major issues. The methodology is appropriate and the data are presented in a well organised way. The analysis is straightforward. The paper adds to the body of work reporting VOC composition from fires in a non-laboratory setting, and should be especially useful to operators of FTIR. I have only a few comments and suggestions listed below. Supplementary materials were referenced in the manuscript but are not accessible online. The paper would benefit from

a tighter definition of the specific fire process being targeted, especially the distinction between pyrolysis and the flaming stage. I don't have a good sense of how important the pre-flame pyrolysis emissions are compared to total emissions from a fire. Do the authors have a metric of what fraction of VOCs (or OH reactivity) is emitted in this process?

Author Comments: We thank the referee for their suggestions to improve the introduction of the paper, specifically the importance between pyrolysis and flaming stages. Also due to the suggestions of the 2nd referee, the introduction is now being heavily revised. We have made changes to the manuscript to reflect the distinction of the phases and the target fire process. With regards to the referee's comment about the supplemental section, the paper has been revised such that no supplemental material is called out in the body of the paper.

Referee Comments: Line 220: What is the OPAG CO detection limit?

Author Comments: We thank the referee for their inquiry of the OPAG's detection limit for CO. Unfortunately, the detection limit for CO measured at 2 cm-1 resolution was approximately 100-500 ppm for a ca. one minute average in a non-optimized configuration. This was due to the moderate CO band strength coupled with interferences from N2O and CO2, but also due to alignment issues and light scattering due to smoke.

Referee Comments: Section 3.1: Since many VOCs were measured during this study, a more robust method of estimating high vs. low pyrolysis fraction could be to fit the entire VOC profiles in ppb provided in Sekimoto et al. 2018 to the measured species. It could also be helpful to compare the overall VOC profile measured here to the high-temperature pyrolysis profile.

Author Comments: We appreciate the referee's comments regarding VOC profiles and high temperature pyrolysis. However, while the IR technique described here identified five new compounds via IR for the first time, including five new VOCs, we believe the IR method provides neither a sufficient number of species nor an extended time profile so

as to provide a broad enough data set from which high and low temperature fractions can be accurately estimated. We suggest combining the IR data with other techniques, e.g. GC and PTR-MS to derive a more expanded profile and more meaningful profile analysis. Such an analysis is beyond the scope of the present paper.

Referee Comments: Table 2: MCE is not reported in the table although it is in the table title.

Author Comments: Per the referee's suggestion, we have updated the table title accordingly.

Referee Comments: Figure 3: The reported mixing ratio (parts-per-thousand) seem excessively high, have similarly high mixing ratios ever been reported previously in the outdoor environment?

Author Comments: The reviewer is correct that most of the mixing ratios seen in Figure 3 are indeed high values, but only three of them actually achieve the parts-per-thousand level. Still, we agree the values are high but they are all for unoxidized hydrocarbon species and we think this is part of what makes this paper unique in that we have captured several hydrocarbon species whose high concentrations may be indicative of the early phases of burning, pyrolysis in particular.

Referee Comments: Figure 5: I'm not sure this figure adds anything to the paper and it could be removed. What do the colours indicate?

Author Comments: We thank the referee for this comment and are happy to clarify. The referee is completely correct in that the data in Figure 5 are somewhat redundant to the data in Table 2. Moreover, the color coded bars are basically superfluous but are added only for (gratuitous?) differentiation. However, we believe the figure does add value for the following reasons: (1) We think it wise to retain the figure to include the molecular structures of the effluents as many of the readers may be non-chemists and as such the structures could be useful; (2) To clearly show that it is the mostly the

lighter alcohols and aldehydes that are predominant species found in the early stage (pyrolysis) emissions, vis-à-vis the aromatics and carboxylic acids which have lower emissions ratios, and (3) the visualization of the standard deviation does graphically emphasize the relatively large uncertainties.

Referee Comments: Line 484: What is meant by secondary reactions? Photolysis or reaction with OH? Or further pyrolysis reaction? Gilman et al. 2015 and Koss et al. 2018 are laboratory studies, so atmospheric reaction seems unlikely. What reaction could reduce the level of naphthalene a factor of 10 on the timescale of a laboratory study? Or is the reduction in ER an effect of elevated CO in the other studies?

Author Comments: We thank the referee for these comments. To clarify, what was meant here are not photolysis or OH-oxidation reactions, but rather additional ring-forming (Diels-Alder) type reactions that lead to larger polyaromatics e.g. anthracene and the like.

---

## Author Response (AR1)

Responses to Referee Comments for ACP 2019

Referee #2

**Referee Comments:** This is interesting work attempting to identify and quantify emissions specifically from the "pyrolysis process" preceding combustion from a prescribed fire. Trace gases were sampled using canisters and quantified offline by FTIR. Emission ratios and emission factors were calculated and compared to a few previous publications. The authors use Sekimoto et al. 2018's ratio of ethyne to furan along with reporting higher ERs for several species compared to other studies as evidence that there was selective sampling of high-temperature pyrolysis emissions. Altogether, this is insufficient evidence that pyrolysis only emissions were selectively captured by the custom sampling device. It is unclear how this body of work is useful to modelers as pyrolysis products are only part of the total fire emissions necessary to accurately model fire. While it does little to inform us about best practices for prescribed burn conditions, it is an interesting attempt to isolate pyrolysis compounds in an "uncontrolled" real-world setting, however, there are entire journals (e.g. Journal of Analytical and Applied Pyrolysis, Fuel, etc.) dedicated to identifying the chemical products from pyrolysis. There is insufficient motivation for why this data is useful and the authors have not made a strong case for publication. It is likely these partial fire results have a strong potential to mislead or confuse the casual reader.

**\*Author Comments:** We thank the referee for the comment, but respectfully disagree with certain aspects of the comment. Along with the first referee and the editor, we believe that this paper represents one of the first, if not the seminal paper, to report field measurements of largely pyrolysis emissions. The referee is correct that there are many other studies of pyrolysis in other journals, but nearly all those studies are controlled laboratory experiments, often in oxygen-free environments utilizing ground (powdered) fuel samples in order to minimize heat transfer effects. This paper is devoted to much more challenging field experiments and should be published as such. It is part of a larger pyrolysis study wherein pyrolysis products from the same plant species are being measured in 1) oxygen-free environment using intact foliage samples, 2) in an atmospheric oxygen wind tunnel setting with relatively simple heterogeneous fuel beds, and 3) in small field burns. One of the goals of the larger study is to determine the relationship between the controlled lab results and actual fire conditions in the field for pyrolysis as was done previously by Yokelson et al. (2013) largely for the combustion and smoldering phases. Results from our objectives 1) and 2) have been published in the open journals and further context to put the present manuscript in perspective have been added. The referee's comment that the gases sampled via our method are not strictly pyrolysis gases is entirely correct. We have mentioned repeatedly in the text that our methods were an imperfect set of "molecular tweezers" to try to capture those pyrolysis species. As discussed in the paper, we believe that we are capturing a much larger fraction of pyrolysis and early-phase species as compared to capturing post-combustion emissions.

**Referee Comments:** In the introduction, a lot of emphasis is placed on the importance of identifying and quantifying biomass burning emissions for model predictions, which guide best practices for prescribed burning conditions. Additionally, the authors highlight the complexities of fires and emission dependencies on fuel types, burn techniques, geometry, etc., however, there isn't proper support/justification for why measurements of just the pyrolysis emissions is vital for these sorts of model predictions or for analyzing burn techniques. A more "realistic" set of emission factors is obtained following combustion as many of the products measured during "pyrolysis only" will likely be consumed and/or combusted in the flame at higher temperatures resulting in different chemical products.

**\*Author Comments:** The referee is correct that smoke composition values for smoke gathered after the onset of combustion are the more useful data for downwind atmospheric models. The focus of the present work, however, is to provide understanding as well as data for physics-based models of the combustion and fire spread process. These models, such as FIRETEC and FDS, often model the process of pyrolysis based on results for wood or on ground foliage samples. An improved modeling of pyrolysis in these models provides an improved suite of chemical species which are then combusted in the models to estimate heat release rates and products of combustion. The focus of the present paper is to better understand the chemistry and physics of the very early stages of the burning process, knowledge which can in an indirect way contribute to realistic chemical models. Still, we have modified the text accordingly so as not to mislead the reader that early stage fire product studies will contribute directly to those models.

**Referee Comments:** The introduction is quite long and needs to remain more focused. It covers everything including downwind formation measured by FTIR to a significant discussion of MCE. It may be useful to rearrange and focus the introduction on motivation and important background considerations of FTIR sampling and pyrolysis measurements and move MCE discussions later in the manuscript.

**\*Author Comments:** We thank the referee for this comment; we have taken this suggestion to heart. The MCE section was moved to the discussion and the entire introduction has been substantially revised.

**Referee Comments:** P4 L 80 There is some discussion in the introduction about the importance of prescribed burning and here the authors reference the necessity to understand burn conditions to minimize gases released during prescribed burns. It may be persuasive to highlight Liu et al., 2017, which provides clear evidence that the PM emitted from a wildfire is much greater than that for a controlled prescribed burn. While the authors have steered clear of PM discussion, this still highlights some major differences observed between the two. It may also be useful to highlight some of the differences between a prescribed burn versus a wildfire in terms of fuel consumption as well as considerations on meteorological and fuel moisture conditions. Liu, X., et al. (2017), Airborne measurements of western U.S. wildfire emissions: Comparison with prescribed burning and air quality implications, J. Geophys. Res. Atmos., 122, 6108– 6129, doi:10.1002/2016JD026315

**\*Author Comments:** We thank the referee for their comments, and his/her knowledge of the literature. We have modified the text to directly reflect these comments and added the citations.

**Referee Comments:** P4 L 89 It is incorrectly stated that pyrolysis is the "first step" in the burning process. Initially there is evaporation of water and other gases absorbed to the solid fuel (distillation), then once the temperature gets high enough to break bonds in the solid fuel both organic aerosol and non-methane organic gases are given off (pyrolysis) and this flammable mixture serves as the airborne fuel that can be rapidly oxidized. Some of the pyrolysis products are processed in the flame to $CO_2$ and BC, however, much of it escapes unoxidized. Entrainments of unoxidized pyrolysis products in smoke naturally falls within definition of smoldering combustion (McKenzie et al., 1995). It might be useful to provide some detail and description of these various processes. It is also important to note that "smoldering" is a complex mixture of multiple processes. For instance, P16 L 308 the authors suggest they aren't measuring products from smoldering combustion, however, distillation, glowing, and pyrolysis are often grouped into smoldering combustion, therefore where is the evidence you are measuring pyrolysis only? McKenzie, L. M., W. M. Hao, G. N. Richards, and D. E. Ward, Measurement and modeling of air toxins from smoldering combustion of biomass, Environ. Sci. Technol., 29, 2047 – 2054, 1995.

**\*Author Comments:** We thank the referee for the comment, and we have modified the text to acknowledge that pyrolysis is not "the" first step, but is one of the first steps.  As to the second comment, "how do we know the measured gases are pyrolysis only?"  The answer to that is we don't know that the gases are only pyrolysis gases.  What we do know and have tried to explain throughout the course of the manuscript is that our imperfect sampling method served as a crude "molecular tweezers" to capture many of the gases prior to onset of combustion.  We note the imperfection of the method in the introduction (in the analysis and discussion sections as well) and point out that it does not capture only pyrolysis gases.  Still, this paper, to the best of our knowledge, is one of the first field experiments to focus on "early-stage" gas-phase emissions.

**Referee Comments:** The goal was to sample before the flame front, though it is likely that glowing combustion is also hot enough to drive pyrolysis. Additionally, the high heat release rate associated with flaming combustion likely pyrolyzes more fuels and there is often a peak in pyrolysis products before, during, or shortly after the peak in flaming product.  Here the authors are likely probing only a portion of the pyrolysis products as sampling preceded the flame front only. There was no attempt to sample pyrolysis products escaping oxidation in the cool flame interiors or following the flame front. Thus, the "pyrolysis products" measured are likely biased/limited to products specifically sampled during the flame front sampling, and it has been shown in other work that pyrolysis mostly occurs after the flame front driven by the heat of glowing (Yokelson et al. 1996). For these reasons, how representative are the pyrolysis emissions measured here?

Yokelson, R. J., D. W. T. Griffith, and D. E. Ward, Open-path Fourier transform infrared studies of large-scale laboratory biomass fires, J. Geophys Res., 101, 21,067 – 21,080, 1996.

**\*Author Comments:** The referee is correct in that pyrolysis gases are not associated strictly with the pre-flame front of the fire, but also occur in the flame envelope and after the flame front. As to how well the present measurements represent (largely) pyrolysis gases is worthy of discussion and indeed is open to the interpretation of the reader.  What we have attempted to do is capture those (pyrolysis) gases prior to the onset of combustion, as there have been many prior studies focusing on either the combustion or smoldering phases (or both) and few or no field studies focusing on the pyrolysis or other early phases. Further studies to better distinguish / characterize the different early phases of the fire, including pyrolysis, are worthy of research by the atmospheric community.

**Referee Comments:** It doesn't appear as though there is any temperature data available, thus how can the authors distinguish between "low-temperature" and "high temperature" pyrolysis? The assignment is purely qualitative. Figure 1 only shows temperatures up to 170C. We'd expect in an oxygen-rich environment the biomass is readily ignited around 500- 600C, therefore most "high temperature" pyrolysis experiments are performed in a N2- atmosphere and more accurately report pyrolysis specific compounds. Determining and quantifying pyrolysis products across a range of temperatures and fuels would have been more useful by heating fuels in a laboratory where the FTIR could instead sample directly.

**\*Author Comments:** Closer inspection of Figure 1 will show that there are in fact several spots in the color coding indicating bright yellow during the flame, and while perhaps difficult to read, the temperature scale at right thus indicates a maximal temperature of 477 °C i.e. nearly 500 °C. We have modified the figure to make the temperature scale more legible in this field experiment.

**Referee Comments:** There is some discussion about spectral regions used and MALT conditions, however, there is no discussion concerning the uncertainty and LODs of your measurements. While much of this may be discussed in Scharko et al. 2018, there still needs to be some information provided in this manuscript.

**\*Author Comments:** The present manuscript is already very long, and in the interest of brevity we still feel it appropriate to direct the reader to the Scharko et al. paper for those discussions.

**Referee Comments:** The detection limits of 12 ppm for acetaldehyde described in the companion paper on this technique are not sensitive enough for most real-world smoke. Do the high LODs of other compounds prevent their detection in this work?

**\*Author Comments:** It may be that the limits of detection for a few of the compounds are sufficiently high that they do indeed preclude their detection via FTIR; we note that acetaldehyde does not have particularly favorable detection limits due to its strongest band near 1750 cm$^{-1}$ being obfuscated by water and other analytes; it was quantified using the weaker C-H stretching lines near 2716 cm$^{-1}$ and thus sensitive detection for this species is more challenging.

**Referee Comments:** Similar to whole air sampling canister-based measurements, there needs to be some discussion about the losses of compounds to the walls of the canister, compounds reacting away, as well as gases formed. Could this account for the higher ERs of alkenes such as ethyne? There is a brief mention of wall losses of NH3, and I suspect this is the major reason no ammonia was measured in this study. The authors incorrectly state that the absence of NH3 supports the fact that low-temperature pyrolysis and/or smoldering isn't measured in this study. It would be useful to cite other FTIR measurements and discussions of ammonia wall losses as well as the potential wall losses of other "sticky compounds" (Stockwell et al., 2014; Yokelson et al., 2003). Yokelson, R. J., Christian, T. J., Bertschi, I. T., and Hao, W. M.: Evaluation of adsorption effects 10 on measurements of ammonia, acetic acid, and methanol, J. Geophys. Res., 108, 4649, doi:10.1029/2003JD003549, 2003.

**\*Author Comments:** We have addressed the referee's valid and well-known concern about the losses of certain compounds reacting away but especially adhering to the walls of the sampling containers in section 3.6, and also added the suggested references.

**Referee Comments:** How useful is calculating an emission factor, considering the definition of EF is the amount of a compound emitted per unit of dry fuel consumed? If measurements only probe the beginning of the process before any combustion has occurred, then the fuel has not been completely consumed. Does this complicate the use of EFs and the assumptions about the amount of carbon consumed considered in the emission factor calculation?

**\*Author Comments:** We address the limited utility of EF in the text. The calculation of the EF is indeed of limited scope here for these and other reasons. We have therefore mostly reported ER values.

**Referee Comments:** P13 L 249 – The authors mention the atmospheric chemistry effects of "total gases emitted during the burns" being the motivation for calculating OH reactivities, however, this is not really an accurate statement as this manuscript is only measuring pyrolysis products preceding combustion. More realistic EFs would be calculated from fire- integrated EFs. Is there other motivation for including a section in the manuscript on OH reactivity?

**\*Author Comments:** Based on this referee's suggestion and other individuals' comments we have decided to remove the OH reactivity and discussion sections from the main manuscript.

**Referee Comments:** Eqn 4: Sekimoto et al., 2018 hasn't shown how effective the use of these two proxies are for real-world fires and the equation was designed to predict total VOC emissions. Using this equation to justify sampling of "high temperature" pyrolysis is not supported. Additionally, in Figure 2 how have the authors decided to compare to these particular studies? Would it be more useful to compare to other field studies, instead of using so many laboratory fire-integrated emissions? It may be useful to focus on field studies with similar fuel types or similar burn conditions (e.g. Yokelson et al 2013, Akagi et al., 2013 "late EFs" calculated for smoldering combustions, Bertschi et al., 2003 samples residual smoldering combustion).

**\*Author Comments:** The fit of the present data to equation 4 and the comparison to the other (fire averaged) studies as seen in Figure 2 was aimed to emphasize precisely that the earlier capture of the (pyrolysis) gases resulted in a very different acetylene/furan ratios.

**Referee Comments:** How did the authors ensure they weren't getting flaming products from fire-driven flow? C2H2 has been observed to increase with increasing MCE (Burling et al., 2011; Yokelson et al 2008) and it's been noted as a "flaming" compound previously. I understand it can also be produced from smoldering combustion, but what makes the author's certain they are capturing pyrolysis produced C2H2?

**\*Author Comments:** As addressed above –we are not certain that there is only pyrolysis-produced $C_2H_2$, there may well be "flaming phase" $C_2H_2$ captured in our samples as well. Although not discussed in this paper due to length considerations, some of the co-authors on this paper also collected gas samples in canisters from both the flaming and pyrolysis stages and analyzed the canisters using GC; the GC results also showed enhanced ratios of $H_2$ and $CO/CO_2$ for samples collected prior to the flame front (i.e. the pyrolysis stage) compared to the gas samples collected during the flaming stage; these measurements provide further evidence that the sampling technique is capturing the gases at a different stage (earlier stage) in the fire.

**Referee Comments:** It is unclear how useful some of the tables and figures are. The comparisons to previous work seems arbitrary. Some justification for how studies were selected for comparisons are needed.

*\*Author Comments:* This comment is rather vague, but we have attempted to respond to the referee's comment; we (and referee #1) believe that the tables and figures as a whole are useful. Also, as noted above, we compared to other (fire averaged) studies, e.g. in Figure 2, to emphasize that our technique resulted in very different acetylene/furan ratios. For example, one of the comparisons included a field study involving prescribed burns at the same site in South Carolina.

Technical corrections:

P2 L 39 Should be "biomass burning emissions"

*\*Author Comments:* The manuscript has been changed to reflect the suggestion.

P3 L 54 More up-to-date citations could be helpful here

**Author Comments:** We thank this referee as well as referee #1 for the same comment. More up to date citations have been added throughout the introduction.

P3 L 71 Similar to specific comment (2): There is some detailed discussion about the downwind changes in fire emissions measured by FTIR. It's not entirely clear why this is relevant in the discussion as no downwind smoke is measured in this study.

*\*Author Comments:* We thank the referee for the comment; we have included the discussions about earlier field studies in order to demonstrate how the present work is different from previous studies and our emphasis on collecting gases prior to arrival of the flame front.

P4 L 84 & L 86 I don't think "hotter" and "cooler" is proper terminology. There is quite a range in temperatures in a fire, both in the flame itself and throughout the fuel bed, thus hotter/cooler are relative terms.

*\*Author Comments:* These adjectives have been dropped.

P4 L 86-87 It might be worth mentioning more OVOCs from smoldering combustion at lower MCE

*\*Author Comments:* We have modified the text accordingly.

P6 L 126 "attempts to ensure" Please change this phrasing, it is confusing.

*\*Author Comments:* Sentence changed to "attempts to collect only gases in front of the flame".

P10 L 182 I don't understand the "aliquots"- Does this mean samples in the same canister were all taken from different areas at different times?

**\*Author Comments:** What we are trying to express is that samples were collected over a short period of time (and area) while keeping the probe in front of the flame to achieve the desired pressure of 138 kPa.

P11 L 204 I'm confused about your H2O and CO2, this wording implies that you did not actually measure these species, though I'm assuming that instead the authors used different spectral regions that weren't optically saturated as values are reported for these compounds later on?

**\*Author Comments:** Correct: We stated that in order to determine the best associated values we did not use those spectral domains that demonstrated saturated absorptions. See line 220-223

P11 L 212- Instead of relative to any "known gas" it should be relative to a co-emitted, long-lived species. Some discussion to describe that CO and CO2 can be generated during glowing, pyrolysis, and flaming would be useful.

**\*Author Comments:** We have changed the wording from "known" to "co-emitted, long-lived" gas.

P11 L 216- The background levels of both CO and CO2 likely changed as fuels were burned in the area. I noticed several burns occurred in time fairly close together, but it is not clear whether the authors took into account the possibility of a changing background or if a constant background was assumed. I believe it is the latter, but this likely adds uncertainty to the ERs.

**\*Author Comments:** This is perhaps true, but likely a minimal effect. First, on all three days there was a period of a few hours between fires that helped dissipate the CO and $CO_2$ levels. Second, on two of the three days, while in close general proximity the second fire of the day was burned at an upwind location from the first fire.

P11 L224- The authors state these are discrete ERs, however, it seems as though there are several samples in the same canister taken from various areas at various times, "aliquots" as they are described. Because of these sampling preferences, is this really a "discrete" sample and how comparable are these to other more traditional grab "snapshot" samples generally seen when sampling with canisters?

**Author Comments:** As already stated, the samples were collected over a short period of time and area while keeping the probe in front of the flame and near vegetation. This collection is different than other studies since the probe is always positioned ahead of the flame front and is not collecting an average over all the stages of the fire; instead the gases are collected near a plant species before the flame has reached it. The use of the word "discrete" is to highlight this difference in which we are trying to only capture gases from one stage of the burn, .i.e. prior to the onset of combustion; of course, as already stated, contributions from flaming emissions may also be present albeit at lower levels.

P12 L 230 also reference Ward and Radke, 1993 Ward, D. E., and L. F. Radke, Emissions measurements from vegetation fires: A comparative evaluation of methods and results, in Fire in the Environment: The Ecological, Atmospheric and Climatic Importance of Vegetation Fires, edited by P.J. Crutzen and J. G. Goldammer, pp. 53 – 76, John Wiley, New York, 1993

**Author Comments:** We thank the referee and have added the reference.

P12 L 243- Stockwell et al. 2015 found these underestimates can be higher for certain fuel types Stockwell, C. E., Veres, P. R., Williams, J., and Yokelson, R. J.: Characterization of biomass burning emissions from cooking fires, peat, crop residue, and other fuels with high-resolution proton-transfer-reaction time-of-flight mass spectrometry, Atmos. Chem. Phys., 15, 845–865, https://doi.org/10.5194/acp-15845-2015, 2015

**Author Comments:** We thank the referee for this comment; we have modified the text and have added the citation.

P13 L 260 This first sentence is not entirely accurate. The combustion efficiency equation is designated to describe combustion chemistry and Sekimoto shows that MCE doesn't necessarily describe their VOC emission profiles. It is true that MCE does not correlate with products as well for pure smoldering, especially for grab samples or in real time, because the same MCE may characterize different white smoke/glowing ratios. MCE is most useful when there is a range of flaming/smoldering as is common in most real-world fires, however, this is not the case in this manuscript.

**\*Author Comments:** We would argue that Sekimoto's description is relevant to the text, namely that the CE or MCE is really not the best metric, as it is a measure of the degree of combustion (hence the name) and the gases that we measure (as for pyrolysis gases as Sekimoto also suggests), are captured before the onset of combustion, rendering the MCE a "less appropriate" metric.

P16 L308- The authors state: "the lower MCE values do not represent the fire burning in the smoldering stage" This is an unsupported claim and as I mentioned earlier smoldering is a grab bag for various processes that include distillation, glowing, and pyrolysis.

**\*Author Comments:** We have partially modified this sentence and point out that the latter part of the sentence clarifies: "but rather suggest that pyrolysis products were captured (at least in part) prior to the onset of combustion."

P16 L 311 This may be a good area to mention fire-driven flow P17

**\*Author Comments:** We are not sure how to interpret this comment. We have added entrainment as a mechanism to incorporate more atmospheric air into the region of sampling as well as the challenge of capturing gases of a moving target.

Figure 2: where are the study average MCEs?

**\*Author Comments:** As discussed, the MCEs are of lesser utility to his study of early-phase non-combustion processes. The study average MCE is found in the title of Table 2. The geometric mean is the most appropriate (of arithmetic, geometric and harmonic) measure of central tendency for ratios such as MCE (Clark-Carter, 2005).

P18 L 336- What was the range of $CO_2$ emissions? If only a narrow spread, this might explain the lower correlation coefficients for something like C2H2 with $CO_2$.

**Author Comments:** The range of $CO_2$ emissions are found in the supplemental material, and generally represent a moderate spread of values.

Table 3. Stockwell et al 2015 has a more complete list of compounds measured including acetaldehyde, acetone, etc. Stockwell, C. E., Veres, P. R., Williams, J., and Yokelson, R. J.: Characterization of biomass burning emissions from cooking fires, peat, crop residue, and other fuels with high-resolution proton-transfer-reaction time-of-flight mass spectrometry, Atmos. Chem. Phys., 15, 845-865, https://doi.org/10.5194/acp- 15-845-2015, 2015.

**\*Author Comments:** This paper has been cited, along with 24 others by Yokelson et al.

P29 L 531 The fuel type and N-composition certainly influences HCN emissions, however, it is random to highlight Indonesian peat here, this is a very unique fuel type and peat burns by smoldering combustion only, thus it follows that the fuel N is biased towards HCN rather than flaming products such as $NO_x$. Not sure it is worth mentioning in this manuscript.

**\*Author Comments:** It is indeed somewhat random to reference the values from Indonesian peat measurements, but the Indonesian peat values are truly are unique/anomalous and are thus worthy of the fleeting reference.

P31 L 555 the authors argue $NH_3$ is only present in the smoldering phase. I'd argue you did measure some smoldering combustion and the most likely for no NH3 detection is wall losses in the canisters as I discussed in specific comment (9).

**\*Author Comments:** This was addressed above. However, we also have OPAG and other data where we clearly do see $NH_3$ in the smoldering stage but not in the earlier stages. This does not mean that wall loss adhesion problems do not exist, but other data does show reduced ammonia and amines in the earlier stages relative to smoldering. These studies will be published in the near future.

The text that has been changed in seen in the revised manuscript below in red font.

[revised manuscript text omitted]

---

## Author Response (AR2)

Comments to the Author:

Thanks for your attention to the reviewers' comments. There are still some issues and unanswered questions that I would ask you to attend to.

You did not answer Reviewer #1's question about what fraction of OH reactivity is represented by these pyrolysis emissions.

Reviewer #1's question, "I don't have a good sense of how important the pre-flame pyrolysis emissions are compared to total emissions from a fire. Do the authors have a metric of what fraction of VOCs (or OH reactivity) is emitted in this process?"

- This is a good question. The fraction of VOCs (or OH reactivity) represented by pre-flame pyrolysis emissions compared to total emissions from a fire can be determined if the total emissions are also measured. Given the present measurements exclusively sampled pre-flame pyrolysis emissions (and not total emissions), the fraction of VOCs (or OH reactivity) cannot be calculated for this particular data set. That said, we have explored this question further using data available in the literature to give the reviewer a sense of the proportion of VOC emitted via pre-flame pyrolysis. It should be noted that the pre-flame pyrolysis period may be quite short in length compared to the other stages of a fire, all of which depend on fuel type and other variables. Using the time profile VOC emissions (including nitrogen and oxygen containing VOCs) provided in the SI by Akagi et al. (2014), the fraction of pre-flame pyrolysis emissions was estimated using measurements from Block 22b and assuming that the time period from 13:40:48–13:45:31 associates with pre-flame emissions (VOC sum of 2.21 ppm) and 13:40:48–16:48:03 for total emissions (VOC sum of 69.8 ppm). The fraction obtained is 0.032, which suggests that the emissions represented by pre-flame pyrolysis are relatively low compared to total emissions. In the study by Akagi et al. (2014), the OPAG was not orientated in such a way as to optimally distinguish and detect pre-flame pyrolysis emissions from other emissions (in that study, the OPAG was specifically set up to capture downwind smoke). For the present manuscript, language has been included suggesting that emissions via pre-flame pyrolysis may be relatively low compared to total emissions from a fire.

Figure 5 is redundant and does not add to the discussion, it should be eliminated.

- The editor is correct. Figure 5 and the associated text have been deleted and remaining text has been reworked.

It seems clear that NH3 will be lost in the analytical system, but the authors are incorrect that other nitrogen species could be lost too, (line 525 revised text). In fact HCN and CH3CN are quite well transmitted by stainless steel sampling components. But the authors are correct that amines will also be readily lost in their system.

- The editor is correct that it is really only NH3 and the amines that are susceptible to this (notorious) sticking problem in such sampling devices. We have revised the text to better reflect the sticking being limited to only NH3 and its amine cronies.

The authors quote the arithmetic mean is given in Table 2, but the Authors' reply says the geometric mean is the most appropriate - why was it not given instead?

- Indeed, after several internal discussions we report only the arithmetic mean in the manuscript. This is because the geometric mean is far less intuitive and understood by fewer folks, including many of the authors on this paper. It would require a few extra sentences to explain and since the paper is already too long we left it out.

The emphasis on peat content is still in the manuscript (line 548), it doesn't seem like these fuels have any peat content. What the authors' did not explain is that the magnitude of nitrogen compound emissions depends on the N-content in fuels, and did not connect their emission measurements with the fuel nitrogen measurements - are there dependencies there that shown up in the data?

- The editor raises an important point about the fuel content, and any correlations between the fuel nitrogen measurements and gas phase emissions. For the N-species it would be especially challenging as both NO and NO2 are largely flaming species, and both are partially obscured by water lines in FTIR analysis, so our NOx data is somewhat lacking here. We do have fairly extensive fuel characterization data coming from our USFS partners, but these data have not been work up yet, and such a study could involve many months of research. It is planned to investigate the fuel content v. emission profiles in an upcoming but separate paper.

Lines 62-66. These two sentences seem at odds with each other.

- Good point. This sentence has been revised.

Line 238. It seems in appropriate to call these ERs discrete since they are averages of a number of aliquots. The sampling scheme is intended to one set of combustion conditions, but the samples are not really discrete.

- Good point. This sentence has been revised.

Lines 312-315. This is a virtual repeat of material in lines 305-308.

- This is indeed redundant and has been eliminated.

Lines 416-418. This material is repetitious, and Figure 5 does not add anything new to the discussion, it can be eliminated and the text tightened up.

- The editor is correct. Figure 5 and the associated text have been deleted and remaining text has been reworked.

Line 421. How does point d) lead to this effect. Doesn't calibration account for this? Or are you saying you are missing entire compounds or classes because they are not detected?

- We have revised the text to indicate that the latter point that the ER relative to other OVOC for e.g. acetaldehyde may be biased high by the fact that other species have IR spectral band strengths or IR spectral interferences that make them appear anomalously low.

Line 582-583. I suggest rephrasing this, as it is hard to follow all the negatives.

➤ The sentence has been revised. We work in the Department of Redundancy Department.